# SparsePO: Controlling Preference Alignment of LLMs via Sparse Token Masks

## Abstract

Preference Optimization (PO) has proven an effective step for aligning language models to human-desired behaviors. Current variants, following the offline Direct Preference Optimization objective, have focused on a strict setting where all tokens are contributing signals of KL divergence and rewards to the loss function. However, human preference is not affected by each word in a sequence equally but is often dependent on specific words or phrases, e.g. existence of toxic terms leads to non-preferred responses. Based on this observation, we argue that not all tokens should be weighted equally during PO and propose a flexible objective termed SparsePO, that aims to automatically learn to weight the KL divergence and reward corresponding to each token during PO training. We propose two different variants of weight-masks that can either be derived from the reference model itself or learned on the fly. Notably, our method induces sparsity in the learned masks, allowing the model to learn how to best weight reward and KL divergence contributions at the token level, learning an optimal level of mask sparsity. Extensive experiments on multiple domains, including sentiment control, dialogue, text summarization and text-to-code generation, illustrate that our approach assigns meaningful weights to tokens according to the target task, generates more responses with the desired preference and improves reasoning tasks by up to 2 percentage points compared to other token- and response-level PO methods.

## 1 Introduction

The rise of employing Large Language Models (LLMs) as conversational agents has increased the importance of aligning them with human preferences. Preference Optimization (PO), i.e. the training paradigm that aims to steer models to a desired behavior (typically related to human perception), is considered the last and most important step in the pipeline of LLM training for producing accurate, harmless and controllable models. Reinforcement Learning from Human Feedback (RLHF; Christiano et al. (2017)) was the primary method for obtaining such a behavior. However, due to it's inherent complexity it has been overpowered by Direct Preference Optimization (DPO) (Rafailov et al., 2023), a simpler, offline approach that produces a policy model that fits the preference data without the need for reinforcement learning.

DPO performs at the sequence level, optimizing rewards and measuring KL divergence for complete responses. However, various studies have shown that signals from specific tokens are primarily responsible for learning desired behaviors, both during pre-training (Lin et al., 2024) and preference optimization (Yang et al., 2024). In particular, in domains where the preference is determined by a specific aspect (e.g. sentiment, toxicity) or when the decision relies on certain subsequences (Pal et al., 2024), it is necessary to consider more fine-grained updates. To further illustrate this point, Figure 1 shows that DPO is already learning implicitly to assign different token-level rewards, with higher values on a few tokens with positive/negative polarity (e.g. *pretty*, *weak*). However, noting the various lone tokens with high rewards, DPO's reward distribution seems inconsistent, and we posit that it would benefit from a more explicit signal.

Aligned with prior work, we argue that not all tokens are important in preference optimization. We further propose that in order to have more diverse responses, and flexible optimization, we should allow only certain tokens to be close to the reference model so that the rest are able to grow beyond it–dismissing the need for measuring KL divergence on all tokens. As such, in this work we pro-

Though I saw this movie the second time I watched it and had to watch it again . The acting was pretty good and the script was very clear about the reason why the kids would do this .

Though I saw this movie on cable yesterday ( which may be a spoiler ), I must say I must say the movie 's storyline is weak in the beginning and there really isn 't anything new to take from that either . The fact it seems to have this plot where it uses the original 's has nothing new to take from that either .

Figure 1: Token-level rewards for chosen (top) and rejected (bottom) responses given an input prompt. After a GPT2-Large model is trained with DPO on the IMDB dataset to generate positive movies reviews, these rewards are calculated as the log ratio of token probabilities between policy (DPO) and reference model (original GPT2-Large). Denser values indicate higher probability score assigned to a token by the policy than the reference, implying importance towards that preference.

pose sparse token-level preference optimization (SPARSEPO), a method that learns automatically during training inherently sparse masks over token-level rewards and KL divergences. Approaches that have been developed based on this observation, either use external models to identify important tokens (Yoon et al., 2024) or need to first perform DPO training to select high-rewardable tokens (Yang et al., 2024). Our method targets flexibility, with masks that can be either shared or independent between rewards and KL divergence. In addition, it is not reliant on external models and can be combined with any possible masking method. In this work, we present two masking strategies but any masking over tokens can be used instead.

Our contributions include (1) a flexible framework, termed SparsePO, for weighting token-level reward and KL contributions tailored to the offline preference optimization objective, (2) analyses over the induced masks' sparsity and reward frontier and how they correlate with controlled KL divergence, (3) quantitative and qualitative gains when employing our proposed approach to different domains with explicit or implicit preference indicators.

## 2 METHODOLOGY

### 2.1 PREFERENCE OPTIMIZATION

The purpose of aligning models with human preferences is to steer model behavior to produce human-acceptable responses. To realize that, we assume training data in the form of static, paired preferences. A prompt $x$ is associated with two responses, chosen $y_c$ and rejected $y_r$, so that $y_c$ is preferred over $y_r$ ($y_c \succ y_r | x$), resulting in a dataset $D = \{x^{(i)}, y_c^{(i)}, y_r^{(i)}\}_{i=1}^N$. Such responses and their rankings are typically collected either by humans or automatically from other models (Xu et al., 2024). In PO, we aim to train a model to generate responses closer to $y_c$ than $y_r$.

In the standard Reinforcement from Human Feedback (RLHF) pipeline (Ziegler et al., 2019) this is realized in a sequence of steps. Firstly, we perform supervised fine-tuning on the task for which we would like to learn preferences, to shift the distribution of the language model in-domain with the PO data. Then, a reward model is trained, responsible for assigning a higher score (reward) to chosen responses and lower scores to rejected ones. Given a policy network $\pi$ (i.e., the model that we aim to optimize), responses are sampled and then scored by the reward model. The policy training aims to maximize the rewards associated with chosen responses and minimize those of rejected ones, subject to a KL constraint with a reference model $\pi_{\text{ref}}$. The constraint prevents the policy $\pi$ from deviating too much from the distribution that the reward model has learned, as well as avoids reward hacking. The above process is translated into the following objective.

$$J_\pi = \max_\pi \mathbb{E}_{x \sim D, y \sim \pi(\cdot|x)} \left[ r(x,y) \right] - \beta \, D_{\text{KL}} \left[ \pi(\cdot|x) \| \pi_{\text{ref}}(\cdot|x) \right], \tag{1}$$

where $r(x,y)$ corresponds to the reward for response $y$ given input $x$, $D_{\text{KL}}$ is the Kullback-Leibler Divergence between the policy $\pi(\cdot|x)$ and the reference model $\pi_{\text{ref}}(\cdot|x)$ over response sequences. In practice, the policy and reference models are the same at the beginning of training while the latter remains frozen.

## 2.2 Sparse Preference Optimization

Motivated by the fact that not all tokens are required to infer a preference, and in order to control token-level contributions, we start by converting the previous objective (Equation 1) that operates on the sequence-level to token-level. Based on the work of Zeng et al. (2024) (TDPO), this corresponds to maximizing the following equation:

$$J_\pi = \max_\pi \mathbb{E}_{x \sim D, y^t \sim \pi(\cdot|x, y^{<t})} \left[ A_{\pi_{\text{ref}}}(y^t|x, y^{<t}) \right] - \beta D_{\text{KL}}[\pi(\cdot|x, y^{<t}) || \pi_{\text{ref}}(\cdot|x, y^{<t})] \quad (2)$$

with $A_{\pi_{\text{ref}}}(y^t|x, y^{<t}) \equiv Q_{\pi_{\text{ref}}}(y^t|x, y^{<t}) - V_{\pi_{\text{ref}}}(x, y^{<t})$ being the advantage function for the reference model as the difference between the state-action $Q$ and the state-value function $V$, and $\beta$ being a tunable parameter controlling the deviation from the reference model. Note that here the KL divergence is over the next-token distribution (i.e., vocabulary).

We argue that in order to control the contribution of each token, we can add a weight in front of the token-level KL divergence term, so that not all tokens are forced to stay close to the reference model. We speculate that this will lead to more diverse generation of responses, since only a handful of important tokens that indicate preference will have to be in-distribution.

Thus, we introduce a mask function $m(y^{<t}) \in [0, 1]$, $m(y^{<t}) > \epsilon$ that produces a scalar for each token $y^t$ in a sequence $y$ that measures the amount of token KL participation in the loss function.

$$J_\pi = \max_\pi \mathbb{E}_{x \sim D, y^t \sim \pi(\cdot|x, y^{<t})} \left[ A_{\pi_{\text{ref}}}(y^t|x, y^{<t}) \right] - \beta \, m(y^{<t}) \, D_{\text{KL}}[\pi(\cdot|x, y^{<t}) || \pi_{\text{ref}}(\cdot|x, y^{<t})] \quad (3)$$

Deriving Equation 3, in a similar manner as TDPO, and assuming that the mask is dependent on the reference model alone and on previously seen tokens, $m(y^{<t}) = f_{\pi_{\text{ref}}}(x, y^{<t})$, we end up with the below optimal policy (refer to Appendix A.1 for a detailed solution),

$$\pi^*(y^t|x, y^{<t}) = \frac{1}{Z(x, y^{<t})} \pi_{\text{ref}}(y^t|x, y^{<t}) \exp\left( \frac{1}{\beta \, m(y^{<t})} Q_{\pi_{\text{ref}}}(y^t|x, y^{<t}) \right), \quad (4)$$

where $Z(x, y^{<t})$ is the partition function.

The Bradley-Terry model (Bradley & Terry, 1952) is a popular theoretical formula employed to model the human preference distribution. As it operates on the sequence-level, its equivalent to the token-level is the Regret Preference model as previously proven by Zeng et al. (2024).

$$P_{BT}(y_c > y_r|x) = \sigma\left( \sum_{t=1}^{T_1} \gamma^{t-1} A_\pi(y_c^t|x, y_c^{<t}) - \sum_{t=1}^{T_2} \gamma^{t-1} A_\pi(y_r^t|x, y_r^{<t}) \right). \quad (5)$$

Solving Eq. 4 for $Q_{\text{ref}}$, considering $A \equiv Q - V$ and substituting to Eq. 5, we obtain the final objective, named SparsePO. Our primary difference is that $m$ is dependent on each token effectively weighting both components of the objective (refer to Appendix A.2 for the detailed solution).

$$\mathcal{L}_{\text{SparsePO}} = -\mathbb{E}_{x, y_c, y_r \sim D}[\log \sigma\left( u(x, y_c, y_r) - \delta(x, y_c, y_r) \right)] \quad (6)$$

$$u(x, y_c, y_r) = \beta \sum_{t=1}^{T_1} m_u(y_c^t) \log \frac{\pi^*(y_c^t|x, y_c^{<t})}{\pi_{\text{ref}}(y_c^t|x, y_c^{<t})} - \beta \sum_{t=1}^{T_2} m_u(y_r^t) \log \frac{\pi^*(y_r^t|x, y_r^{<t})}{\pi_{\text{ref}}(y_r^t|x, y_r^{<t})} \quad (7)$$

$$\delta(x, y_c, y_r) = \beta D_{\text{MaskKL}}[x, y_c; \pi^* || \pi_{\text{ref}}] - \beta D_{\text{MaskKL}}[x, y_r; \pi^* || \pi_{\text{ref}}], \quad (8)$$

where $D_{\text{MaskKL}}[x, y; \pi^* || \pi_{\text{ref}}] = \sum_{t=1}^T m_d(y^t) \, D_{\text{KL}}[\pi^*(\cdot|x, y^{<t}) || \pi_{\text{ref}}(\cdot|x, y^{<t})]$. The objective effectively adds token-level masks $m_u$ on rewards (Equation 7) and $m_d$ on the KL (Equation 8) for each response respectively. Naturally, these masks can either be shared or be independent. In the following sections we experiment with both $m_u = m_d$ and $m_u \neq m_d$.

## 2.3 Mask Computation

In the previous section we showed how we can control the contribution of rewards and KL divergence of each token through the introduction of weights in the loss function. Next, we introduce two strategies to obtain these weights from the reference model, one that is derived directly from its internal activations and another that is learned in parallel during preference optimization.

**Model Activation-based Mask (MAPO)**

Inspired by mechanistic interpretability approaches (Huben et al., 2023), we leverage the rich information captured per token in the activations of the reference model and aggregate them into token-level weighting masks, as follows. Let $a_g^t \in R^{d'}$ be the output of activation function $g(*)$ in network $\pi_{ref}$, and $\bar{a}_g^t$ its average value across dimensions for time step $t$. Note that $a_g^t$ is exposed to information from $y^{<t}$ due to the autoregressive nature of generation. We obtain $[\tilde{a}_g^1, .., \tilde{a}_g^T]$, where $\tilde{a}_g^t = (\bar{a}_g^t - \text{mean}(\bar{a}_g))/std(\bar{a}_g)$ is the standardization of $\bar{a}$ across sequence $y$. Then, we define activation-based mask $m(y^{<t}) = \text{mean}\{\tilde{a}_g^t | \forall g \in \pi_{ref}\}$, i.e. the average $\tilde{a}_g^t$ for all activations in the reference model. In practice, we aggregate outputs from feed-forward layers, residual connections, and attention layers, across all layers in $\pi_{ref}$. Finally, we set $m_u(y^{<t}) = m_d(y^{<t}) = m(y^{<t})$, i.e. a common mask for the rewards and KL terms given.

**Learnable Sparse Mask (SPARSEPO)**

In our second variant, mask $m(y^{<t})$ is computed using learnable parameters. Specifically, we learn one feed-forward network (FFN) with ReLU activation for each model layer, and aggregate representations from all layers with a linear layer.[1] A single layer mask is computed as follows:

$$m^{(l)}(y^{<t}) = ReLU\left(\mathbf{H}^{(l)}(y^t) \cdot \mathbf{w}^{(l)} + \mathbf{b}^{(l)}\right),$$

where $\mathbf{H}^{(l)} \in \mathbb{R}^{N \times d}$ corresponds to the reference model hidden representation for layer $l$ for $N$ tokens and $\mathbf{w}^{(l)} \in \mathbb{R}^d, \mathbf{b}^{(l)}$ are the $l$-layer learned parameters. Consequently, when learning multiple masks per layer, they are combined as

$$m(y^{<t}) = ReLU\left(\text{Concat}\left(m^{(1)}(y^{<t}), ..., m^{(L)}(y^{<t})\right) \cdot \mathbf{w}_o\right),$$

with $\mathbf{w}_o \in \mathbb{R}^L$ the output merging vector.

The ReLU activation function produces a sparsity in the masks, the degree of which is dependent on the target preference data and the reference model. The mask values (independent of strategy) are utilized solely during PO training and are ignored during model inference.

## 3 EXPERIMENTS

In this section, the effectiveness of SparsePO is investigated in both proxy-preference and human-preference setups. Proxy-preference setups are analyzed through sentiment control, summarization, and code generation, whereas human-preference setup is analyzed through single-turn dialogue tasks. We refer the reader to Appendix B for further details on experimental setup.

### 3.1 MODEL COMPARISON

We compare the performance of SparsePO against supervised fine-tuning over preferred responses (SFT, serving both as a baseline and the starting point of the PO variants) and performant PO strategies that model preference at the sequence and token levels. At the sequence level, we compare against DPO (Rafailov et al., 2023), which aims to mitigate KL divergence; SimPO (Meng et al., 2024), which aims to maximize the probability difference between chosen and rejected responses; and DPOP (Pal et al., 2024), which adds a penalty term to the DPO loss to encourage high probability scores of the preferred completions. At the token level, we compare against TDPO v1 and v2 (Zeng et al., 2024), which adds token-level KL divergence as a regularization term. Unless stated otherwise, we investigate SparsePO setups learning a common mask for reward and KL terms ($m_u = m_d$) as well as learning different ones ($m_u \neq m_d$).

### 3.2 SENTIMENT CONTROL

Following prior work (Rafailov et al., 2023; Amini et al., 2024; Zeng et al., 2024), we use sentiment as a proxy for preference and align models to generate positive movie reviews. For the SFT model,

---

[1] We initially experimented with learning two FFNs per layer, one for the chosen and one for the rejected responses. However this led to overfitting, hence we learn a single vector per layer.

we use GPT2-LARGE (Radford et al., 2019) trained on the IMDB dataset (Maas et al., 2011).[2] To train PO, preference data is generated by sampling two completions per review prefix from the SFT model. Then, we use a sentiment classifier[3] as a ground-truth reward model and set chosen ($y_c$) and rejected ($y_r$) responses such that score($y_c$) > score($y_r$), where score(y) = $p(y|\text{positive})$ or $1 - p(y|\text{negative})$ if $y$ is classified as positive or negative, respectively.

**Reward and KL Divergence Trade-off.** We start our analysis by investigating the trade-off between ground-truth reward and response-level KL divergence by estimating their Pareto frontier. For all policies, we train using $\beta = \{0.01, 0.1, 0.2, ..., 1, 2, 3, 4, 5, 10, 20\}$ and for SimPO, $\gamma$ in $\{0.3, 1\}$. For each policy variation, we generate one response per prompt in the test set using multinomial sampling, every 100 training steps, and report the the ground-truth reward and the average response-level KL divergence, averaged over samples.

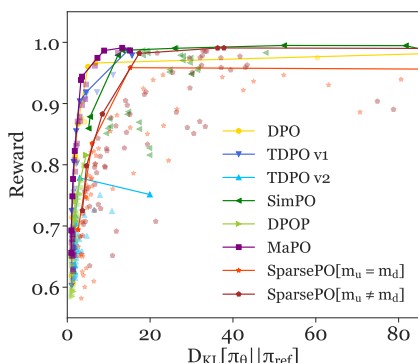

Figure 2: Pareto frontier of expected reward and response-level KL divergence w.r.t. the reference model, for a sentiment control scenario over the IMDB dataset. Solid lines estimate the frontier for each system, and points represent hyper-parameter variations.

The following insights can be gathered from the frontier, showcased in Figure 2. We observe that DPOP restricts KL divergence and reward to under 5 and 0.82, TDPO v1 to 15 and 0.97, TDPO v2 to 19 and 0.75, and SimPO to 81 and 0.99. This shows that TDPO v2 allows slightly larger KL divergence than v1 but it does not reach higher rewards. Among our proposed systems, MaPO notably dominates the frontier, reaching a moderate KL of 15 and a reward of 0.99, higher than DPO (0.96) and comparable to SimPO. On the other hand, SparsePO variants allow a much larger effective KL divergence range, with higher concentration of system points at high KL values than any baseline. Regarding rewards, although the independent mask setup ($m_u \neq m_d$) does reach a reward of 0.99, the common mask setup ($m_u = m_d$) seems to trade off divergence range for a slight decrease in reward (0.95). These results demonstrate that the proposed masking strategies are effective at balancing expected ground-truth reward and response-level KL divergence.

**Sparsity and Token-level KL divergence.** Next, we analyze the trade-off between mask sparsity and token-level KL divergence throughout training, in the independent mask setup of SparsePO. Figure 3 shows results for chosen responses from systems trained at different values of $\beta$.[4] Firstly, we note that sparsity in the reward mask ($m_u$) always starts high (80%), increasing slightly and then steadily decreasing until the end of training, reaching as down as 20%. Such decrease is controlled by increasing $\beta$ until 0.8, after which the trend is inverted. We hypothesize that the reward mask first learns to identify the tokens most informative for sentiment control, and increasingly expands this token set as training proceeds at a rate controllable by $\beta$. This insight adds to previous findings (Yang et al., 2024) stating that PO-trained models can learn to identify highly rewardable tokens.

Regarding the divergence mask, we find that increasingly higher values of $\beta$ induce higher levels of sparsity in $m_d$, restricting the amount of tokens allowed to diverge in a sequence, which translates to lower token-level KL divergence throughout training. However, for sufficiently low values of $\beta$, sparsity can be kept below 20%.

In summary, we find that low values of $\beta$ induce scenarios where reward sparsity is high and divergence sparsity is low, meaning that the loss is dominated by term $\delta(x, y_c, y_r)$. Conversely, a high $\beta$ induces high sparsity on both masks, hindering learning significantly. However, we do observe that a more balanced sparsity level in both masks can be induced with mid-range values of $\beta$.

**Qualitative Analysis.** Finally, we perform qualitative analysis on the learned masks by observing their token-level values on example sentences. Similarly to Figure 1, we calculate token-level rewards as the log ratio of response probabilities between policy and reference models. Token-level

---

[2]https://huggingface.co/insub/gpt2-large-imdb-fine-tuned

[3]https://huggingface.co/siebert/sentiment-roberta-large-english

[4]See Figure 11 in Appendix C for similar plots over rejected responses.

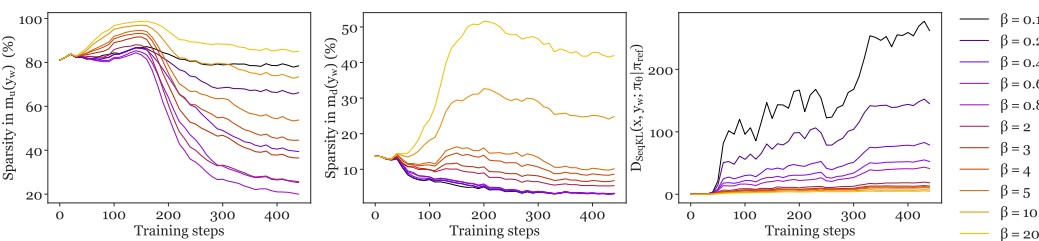

Figure 3: Sparsity levels in the reward mask ($m_u$, left) and the token-level KL divergence mask ($m_d$, middle), as well as token-level KL divergence of *chosen* responses during training (over IMDB), for increasing values of $\beta$.

**TDPO-v2 rewards:** Though I saw this movie the second time I watched it and had to watch it again . The acting was pretty good and the script was very clear about the reason why the kids would do this .

**SparsePO-common rewards:** Though I saw this movie the second time I watched it and had to watch it again . The acting was pretty good and the script was very clear about the reason why the kids would do this

**SparsePO-indp rewards:** Though I saw this movie the second time I watched it and had to watch it again . The acting was pretty good and the script was very clear about the reason why the kids would do this .

(a) Chosen response rewards.

**TDPO-v2 KL:** Though I saw this movie the second time I watched it and had to watch it again . The acting was pretty good and the script was very clear about the reason why the kids would do this .

**SparsePO-common KL:** Though I saw this movie the second time I watched it and had to watch it again . The acting was pretty good and the script was very clear about the reason why the kids would do this .

**SparsePO-indp KL:** Though I saw this movie the second time I watched it and had to watch it again . The acting was pretty good and the script was very clear about the reason why the kids would do this

(b) Chosen response KL values.

Figure 4: Token-level heatmaps for chosen responses for TDPO-v2 SparsePO. Darker color indicates higher values. All scores are scaled in $[0, 1]$ for comparison.

KL divergence is calculated as the token-level KL between policy and reference. We show the values of reward and KL divergence *after* the mask application in a common mask setup($m_u = m_d \rightarrow$ *common*) and on independent setup ($m_u \neq m_d \rightarrow indp$). We also compare with the TDPO baseline as the closest method to ours. Technically, when $m_u = m_d = 1$ our objective becomes equivalent to TDPO, hence we can check the influence of the proposed masks on the target objective. Figure 4a illustrates that a common mask has less sparsity compared to independent, highlighting a larger set of tokens. Comparing directly reward maps with TDPO we see that that independent mask is weighting only subsequences that express a certain polarity (*watch it again*), while TDPO gives a weight to all tokens in the sequence. The same stands for common masks while being slightly noisier in the tokens they cover. Looking at KL divergence maps in Figure 4b, lower values indicate minor to no divergence from the reference model. TDPO is stricter in KL control, forcing the majority of tokens to be close to the reference model, while common and sparse masks allow more diversity with higher values on particular tokens, possibly easing diversity. Heatmaps for the rejected response can be found in Figure 21.

### 3.3 HELPFULNESS & HARMLESSNESS CONTROL

Here, we investigate the effectiveness of our approach in aligning models to generate helpful and harmless responses in dialogue. We employ the Anthropic HH dataset (Bai et al., 2022), consisting of open-ended multi-turn dialogues in which humans ask a chat assistant for help, advice, or to perform a task. We train Pythia 1.4B (Biderman et al., 2023) using the chosen completions for SFT training and the preference dataset for PO over the resulting reference model.

For evaluation, we report performance in reasoning and instruction following tasks over Hugging-Face's OpenLLM Leaderboard v2,[5] and use the LM Evaluation Harness framework (Gao et al.,

---

[5]https://huggingface.co/spaces/open-llm-leaderboard/open_llm_leaderboard

| Methods | BBH | MATH | GPQA | MuSR | MMLU PRO | IFEval Inst. | IFEval Prom. | Avg |
|---|---|---|---|---|---|---|---|---|
| SFT | 2.87 | 0.30 | 0.78 | 4.02 | 1.71 | 25.90 | 14.97 | 7.22 |
| DPO | 2.64 | 0.60 | 0.00 | 3.77 | 1.19 | 21.46 | 10.54 | 5.74 |
| TDPO-v1 | 3.01 | 0.53 | 0.00 | 4.30 | 1.50 | 20.62 | 9.98 | 5.71 |
| TDPO-v2 | 2.65 | 0.23 | 0.00 | 5.87 | **1.68** | 18.47 | 8.32 | 5.32 |
| SimPO | 2.10 | 0.00 | 1.12 | 4.36 | 1.41 | 19.90 | 9.24 | 5.45 |
| DPOP | 2.71 | 0.68 | **1.57** | 3.85 | 1.43 | 20.02 | 9.06 | 5.62 |
| MaPO | 3.60 | **0.91** | 0.00 | 3.94 | 1.33 | **22.78** | **12.57** | 6.45 |
| SparsePO$[m_u = m_d]$ | 3.24 | 0.23 | 0.00 | **6.67** | 1.25 | **22.78** | 12.38 | **6.65** |
| SparsePO$[m_u \neq m_d]$ | **4.10** | 0.76 | 0.00 | 3.45 | 1.42 | **22.78** | 11.28 | 6.25 |

Table 1: Performance of Pythia 1.4B models on Open LLM Leaderboard 2 after PO with Helpfulness & Harmlessness as proxy for human preference. Best number across PO methods are bolded.

2024) for metric calculation. Additionally, we calculate win rates against a baseline policy, using GPT-4 as a proxy for human evaluation of helpfulness and harmlessness.[6] We randomly sample 100 instances among the single-turn dialogue instances in HH. Chosen responses are used as baseline and 5 system completions are sampled per prompt using nucleus sampling with $p = 0.95$ at temperatures $\{0, 0.25, 0.5, 0.75, 1.0\}$.

**Alignment, Reasoning and Verifiable Instruction Following.** In terms of average score, showcased in Table 1, SFT performs better than all systems, indicating a sharp trade-off between alignment objective and task performance, regardless of the PO strategy. This could indicate that by making a model more helpful and harmless, we sacrifice some reasoning capabilities (Luo et al., 2023). Nevertheless, our proposed alignment strategies, MaPO and SparsePO –both at common and independent mask setups– demonstrate their effectiveness at balancing alignment goals and reasoning, being the best among PO strategies.

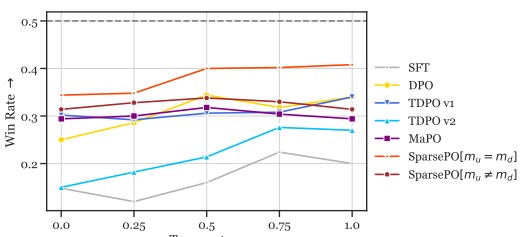

Figure 5: Win rates of system completions against chosen responses in Anthropic HH single-turn dialogue, using GPT-4 (gpt-4-turbo) as a judge.

In terms of specific reasoning and task type, the following can be noted. Firstly, our mask strategies are effective for certain types of reasoning. Although mathematical reasoning (MATH) poses a challenge to all systems, MaPO outperforms all baselines including SFT, followed by SparsePO$[m_u \neq m_d]$. Similarly, SparsePO$[m_u \neq m_d]$ performs best at BBH, followed by MaPO, indicating a better handling of factual and world knowledge as well as algorithmic reasoning. Multi-step soft reasoning tasks (MuSR) are best handled by SparsePO$[m_u = m_d]$, followed by TDPOv2. However, tasks that require extensive knowledge (GPQA and MMLU-pro) pose a challenge to all systems, and our masking strategies in particular. Similarly, tasks based on verifiable instructions (IFEval), both instruction and prompt based, exhibit the starkest trade-off between alignment and task performance, given the sharp decrease in metric scores after preference optimization. Still, SparsePO and MaPO outperform all other PO strategies, trailing second only to SFT. Finally, regarding win-rates, SparsePO surpasses all methods with +6.8% over TDPO-v1, +12.6% over TDPO-v2 and +5.6% over DPO.

## 3.4 Summary Quality Control

In this task, we employ overall summary quality as proxy for human preference, which includes quality aspects such as information coverage, faithfulness, and coherence. We use the Reddit TL;DR dataset (Völske et al., 2017) and its preference annotations (Stiennon et al., 2020) to fine-tune a GPTJ-6B (Wang & Komatsuzaki, 2021) SFT model[7] using LoRA (Hu et al.). Here we only analyze

---

[6]Please refer to Appendix B.4 for details about he prompt used.

[7]https://huggingface.co/CarperAI/openai_summarize_tldr_sft

representative baselines from sequence and token-level preference modeling (DPO, TDPO v1 and v2) against MaPO and SparsePO in common mask setup.

For evaluation, we take 100 prompts from the test set and sample 5 completions using nucleus sampling ($p = 0.95$) and temperatures $T = \{0, 0.25, 0.50, 0.75, 1.0\}$. Regarding automatic metrics, we report ROUGE-L $F_1$ (Lin & Hovy, 2003) and BERTScore $F_1$ (Zhang et al., 2020) for lexical and semantic relevance, respectively; self-BLEU (Zhu et al., 2018) for lexical diversity; and EDNA (Narayan et al., 2022), a metric quantifying diversity and faithfulness by combining document-summary entailment (Laban et al., 2022) and self-entailment. Additionally, similar to the previous section, we report win rates of system summaries against reference summaries using the same prompts and sampled completions mentioned above (prompt available in Appendix B.3).

**Alignment, Diversity, and Faithfulness.** We investigate how our method balances alignment accuracy –as measured by summary relevancy–, generation diversity, and faithfulness. Figure 6 presents metric scores across temperature values, for test set instances with high document–reference summary faithfulness (Aharoni et al., 2023), i.e. $P_{\text{ent}}(D \models S_{\text{ref}}) > 0.6$. Both SparsePO setups achieve comparable relevancy and diversity scores to the baselines, whilst MaPO obtains lower relevancy at $T = \{0.25, 0.5\}$. However, EDNA scores indicate that $\text{SPARSEPO}[m_u = m_d]$ does perform best at $T = 0.25$, and remains competitive at higher temperatures. This shows that, when learning a common mask, SparsePO is able to produce faithful and diverse summaries without trading off relevancy at low temperatures.

In terms of win-rates (see Figure 7), we observe that MaPO is the overall best PO method, achieving comparable performance to others across temperatures, while being marginally better at 0.25 and offering a 6.4% improvement at 1.0. On this domain, sparsity results in suboptimal performance.

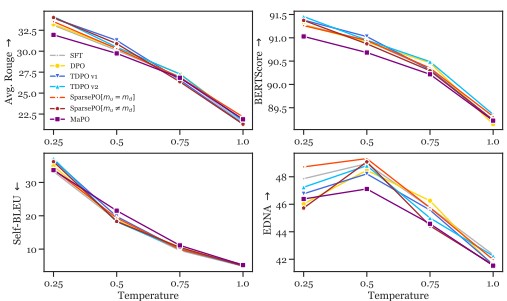

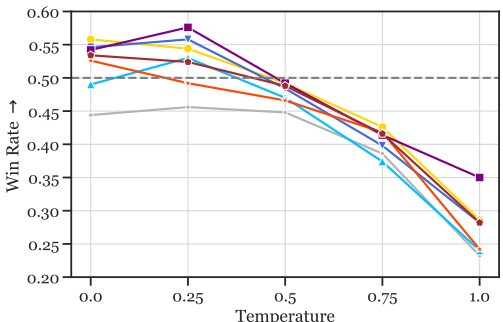

Figure 6: Summary relevancy (avg. ROUGE, BERTScore), lexical diversity (Self-BLEU), and entailment-based diversity and faithfulness (EDNA), over highly faithful instances of the TL;DR test set ($P(D \models S_{\text{ref}}) > 0.6$).

Figure 7: Win rates against reference summaries from the TL;DR test set, using GPT-4 (gpt-4-turbo) as a judge.

### 3.5 TEXT-TO-CODE GENERATION

Finally, we perform preference optimization for the task of text-to-code generation, using a simple preference dataset created from Python programming problems from Gee et al. (2024). In this experiment, we aim to optimize for correctness, i.e., a chosen program is an executionable one that passes all accompanied unit-tests and a rejected program is one with the opposite behavior. The MBPP dataset (Austin et al., 2021) is employed, which consists of 384 train, 90 validation and 500 test programs. We use StarCoder-1B (Li et al., 2023) to sample 100 solutions for each problem in train and validation with multinomial sampling. After testing the generated programs, we end up with 183 prompts with at least two passing and one failed solution for the training set and 40 for the validation set. The preference data is built by selecting randomly different pass-fail solutions for each prompt at every epoch. Using the resulting data, we use StarCoder-1B for PO training. Performance is measured in terms of functional correctness[8] on MBPP and HumanEval (Austin et al., 2021), sampling 100 solutions with temperature 0.6 and $p = 0.95$ in Table 2.

---

[8]A functionally correct response is one that executes and produces the correct answer to all test cases.

| METHOD | HUMANEVAL | | | MBPP | | |
|---|---|---|---|---|---|---|
| | PASS@1 | PASS@10 | PASS@100 | PASS@1 | PASS@10 | PASS@100 |
| STARCODER-1B | 12.22 | 24.69 | 38.41 | 17.83 | 39.94 | 59.60 |
| DPO | **14.61** | **28.42** | **46.34** | 21.36 | **44.71** | 62.40 |
| TDPO-v1 | 14.46 | 27.42 | **46.34** | 21.58 | 44.48 | 61.60 |
| TDPO-v2 | 13.30 | 26.06 | 45.73 | 19.93 | 42.51 | 62.00 |
| SIMPO | 14.55 | 27.74 | 45.73 | **22.89** | 43.63 | 59.20 |
| MAPO | 14.12 | 27.30 | 42.07 | 20.93 | 43.63 | 62.20 |
| SPARSEPO$[m_u = m_d]$ | 14.15 | 27.32 | 42.68 | 20.92 | 44.25 | **64.80** |
| SPARSEPO$[m_u \neq m_d]$ | 14.39 | 28.29 | 44.51 | 19.81 | 43.71 | 62.00 |

Table 2: Pass@$k$ results for text-to-code generation using StarCoder-1B.

Overall, DPO shows the strongest performance across the board on HumanEval for all pass@$k$ setups, while all methods manage to improve over the baseline SFT model. Our proposed models tend to perform on par with other PO methods although worse on pass@100. On MBPP though, SparsePO shows gains over pass@100, offering a $+2\%$ improvement compared to DPO, with a slight decay in the remaining metrics. The discrepancy between HumanEval and MBPP could be attributed to the MBPP being the in-domain PO data.

These results indicate that although SparsePO is weighting more tokens as important for preference, in the code domain and in particular code execution, this requirement cannot be easily satisfied. In fact, code sequences are heavily structured and every 'word' is intricately reliant on all other 'words' in the sequence, i.e. there is little information that may be considered redundant. As such, a weighing scheme (such as in SparsePO) will effectively ignore parts of the sequence that can be crucial; this is further supported from qualitive analysis presented in Figure 22 in the Appendix. Since the goal of the task is to improve functional correctness (whether a programs runs correctly or not) ignoring any 'word' in a code sequence will most certainly lead to a functionally incorrect solution. This is in contrast to natural language, where some words are naturally more important for preference than others. This includes the standard Preference Optimization goals of reducing toxicity or style adaptation, but it extends on reasoning tasks as well when that reasoning is happening through natural language. This also explains SparsePO's benefits to the MATH benchmark, as performance there is enabled by Natural Language instructions through chain-of-thought reasoning.

Similarly to sentiment control, we also report sparsity values as a function of training steps for models trained with different values of $\beta$; see Figures 12 and 13 in the Appendix.

## 4 RELATED WORK

Since the introduction of DPO, several methods have been developed to mitigate the various shortcomings of the method, mostly by introducing further constrains to the loss function. Identity Preference Optimization (Gheshlaghi Azar et al., 2024, IPO) was proposed to primarily tackle overfitting, that does not rely on the Bradley-Terry modulation assumption. Ethayarajh et al. (2024) introduced KTO, that takes advantage of that Kahneman-Tversky model of human utility. The method drops the requirement for preference pairs and is dependent only on a binary signal of whether a response is acceptable or not. To control response length and dismiss the need for a reference model, SimPO (Meng et al., 2024) uses the average log probability of the sequence (instead of the sum) while also requiring the difference between responses to be at least equal to a margin. Another method that does not require a reference model or prior supervised fine-tuning, is ORPO (Hong et al., 2024), that optimizes the odds ratio together with cross-entropy. On a similar vein, Amini et al. (2024) argues that not all preference pairs are considered equal, requiring the preferred responses to have a likelihood larger than an offset value from the dispreferred ones, based on the score assigned to each response from an external reward model. Other methods that incorporate margins between probability differences include DPO-Positive (Pal et al., 2024), where the log probability of the preferred response for the policy needs to be higher than that of the reference model. The method is particularly effective when the edit distance between responses is low, e.g in math data. Wu et al. (2024) specifically aimed at a dynamic optimization of the $\beta$ value for each batch, proposing $\beta$-DPO.

Closer to our approach, there is a family of methods that focus on token-level rather than sequence-level optimization. In TDPO (Zeng et al., 2024), the sequence-level DPO objective is converted into token-level, which results in the KL divergence to act as a regularization term, optimized together with the original objective. The new loss leads to more controllable KL values throughout the course of training. Inverse-Q*(Xia et al., 2024) optimizes the same objective as PPO assigning token-level reward feedback via an estimated policy. Similarly, Token-level Continuous Rewards (Yoon et al., 2024, TLCR) incorporate a discriminator trained to distinguish positive and negative tokens (obtained from GPT-4 judgments). The confidence of the discriminator is used to assign continuous rewards to each token considering the context. Similarly to our motivation, in Selective PO (Yang et al., 2024, SePO), not all tokens are considered equal. An oracle model is trained first to identify which tokens are important in chosen and rejected responses (based on their reward values). These tokens are then used to train DPO again, while the rest are zeroed out. In contrast to the above methods, we aim for maximum flexibility. Our approach does not require an external LLM to model rewards and our proposed masks are learned on the fly, effectively assigning higher rewards to tokens that are important to the target preference. In addition, SparsePO induces the necessary sparsity in the masks automatically with a single stage of training.

## 5 DISCUSSION

Based on the controlled experiments we conducted in the previous section, here we briefly discuss our overall findings. Firstly, based on the sentiment control analysis, SparsePO allows larger KL divergence at little to no cost in expected ground-truth reward. The $\beta$ value is able to control sparsity in both masks, across domains, with values between $0.6$ to $4$ leading to mid-range sparsity levels. Depending on the domain and target preference proxy, we found that higher sparsity was present in sentiment control, highlighting a certain triviality of the task as the SFT model seems able to already identify words that are important for the target preference. On the other end, for code generation and summarization, lower sparsity between $0.2$ and $0.4$ seemed best in terms of alignment accuracy as executability and summary correctness are less well-defined preference proxies. For helpfulness control, optimal sparsity was found instead between $0.6$ and $0.8$, possibly as existence of toxic terms immediately renders response dispreferred. We would argue that the mask works in tandem with beta and we observed that the range of betas that are effective with SparsePO is generally higher than DPO (with best values between 0.4-1).[9]

From our analysis over DPO, TDPO and their variants, it is important to note that, although restricting divergence at the response or token-level proves effective at maintaining the model in-domain, this does not guarantee better ground-truth rewards or better downstream task performance. For cases in which the preference proxy is complex, such as 'helpfulness', 'summary quality' or 'executability', this plain control can even hinder performance. In contrast, we devise a training procedure in which a model can learn to enhance or suppress the reward and KL divergence for each token independently. Our qualitative analysis shows that indeed for trivial tasks tokens important towards the preference get high rewards and low KL divergence, meaning they need to be close to the reference predictions to maintain preference.

## 6 CONCLUSION

We introduced Sparse Token-level Preference Optimization (SparsePO), a novel LM alignment strategy that learns to weight the reward and KL divergence for each particular token in a response during PO training. We proposed two masking strategies, obtaining model activation-based masks from the reference model and learning mask representations either commonly for both reward and divergence terms or independently. By allowing masks to be learned along with preference, we observed that they converged to a non-trivial level of sparsity which can be controlled with well-studied hyper-parameters in preference optimization, while being dependent on target preference proxy. Extensive experiments across several tasks and domains, reveal that our method consistently outperforms strong baselines that model preference at the response and token-level, while assigning higher rewards and lower KL values to tokens that are important for inferring target preference. SparsePO can be easily extended to use other masking strategies and can be combined with other PO variations.

---

[9]Removing $\beta$ ($= 1.0$) results in slightly suboptimal performance.

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

# A   MATHEMATICAL DERIVATIONS

## A.1   OBTAINING THE OPTIMAL POLICY

In order to get the optimal policy, we take advantage of $A(y^t|x, y^{<t}) \equiv Q(y^t|x, y^{<t}) - V(x, y^{<t})$ and solve the following objective that includes our introduced mask $m(y^{<t})$. In the following equations, $\pi$ refers always to next-token distribution $\pi(\cdot|x, y^{<t})$, and we oftentimes omit $(y^t|x, y^{<t})$ for simplicity.

$$J_\pi$$

$$= \max_\pi \mathbb{E}_{x,y^{<t}\sim D, y^t \sim \pi} \left[ A_{\pi_{\text{ref}}}(y^t|x, y^{<t}) \right] - \beta\, m(y^{<t})\, D_{\text{KL}}[\pi(\cdot|x, y^{<t}) || \pi_{\text{ref}}(\cdot|x, y^{<t})]$$

$$= \max_\pi \mathbb{E}_{x,y^{<t}\sim D, y^t \sim \pi} \left( \left( Q_{\pi_{\text{ref}}}(y^t|x, y^{<t}) - V_{\pi_{\text{ref}}}(x, y^{<t}) \right) + \beta\, m(y^{<t})\, \log\left( \frac{\pi_{\text{ref}}(y^t|x, y^{<t})}{\pi(y^t|x, y^{<t})} \right) \right)$$

$$= \max_\pi \beta\, \mathbb{E}_{x,y^{<t}\sim D, y^t \sim \pi} \left( \log e^{\frac{1}{\beta} Q_{\pi_{\text{ref}}}(y^t|x, y^{<t})} - \frac{1}{\beta} V_{\pi_{\text{ref}}}(x, y^{<t}) + \log\left( \frac{\pi_{\text{ref}}(y^t|x, y^{<t})}{\pi(y^t|x, y^{<t})} \right)^{m(y^{<t})} \right)$$

$$= \max_\pi \beta\, \mathbb{E}_{x,y^{<t}\sim D, y^t \sim \pi} \log \left( \frac{\pi_{\text{ref}}^{m(y^{<t})}(y^t|x, y^{<t}) \exp\left( \frac{1}{\beta} Q_{\pi_{\text{ref}}}(y^t|x, y^{<t}) \right)}{\pi^{m(y^{<t})}(y^t|x, y^{<t})} \right) - \frac{1}{\beta} V_{\pi_{\text{ref}}}(x, y^{<t})$$

$$= \max_\pi \beta\, \mathbb{E}_{x,y^{<t}\sim D, y^t \sim \pi} \log \left( \frac{\frac{Z(x,y^{<t})}{Z(x,y^{<t})} \pi_{\text{ref}}^{m(y^{<t})} \exp\left( \frac{1}{\beta} Q_{\pi_{\text{ref}}} \right)}{\pi^{m(y^{<t})}} \right) - \frac{1}{\beta} V_{\pi_{\text{ref}}}$$

$$= \max_\pi \beta\, \mathbb{E}_{x,y^{<t}\sim D, y^t \sim \pi} \log \left( \frac{\frac{1}{Z(x,y^{<t})} \pi_{\text{ref}}^{m(y^{<t})} \exp\left( \frac{1}{\beta} Q_{\pi_{\text{ref}}} \right)}{\pi^{m(y^{<t})}} \right) - \frac{1}{\beta} V_{\pi_{\text{ref}}} + \log Z(x, y^{<t})$$

$$= \max_\pi \beta\, \mathbb{E}_{x,y^{<t}\sim D, y^t \sim \pi} \log \left( \frac{1}{Z(x,y^{<t})} \pi_{\text{ref}}^{m(y^{<t})} \exp\left( \frac{1}{\beta} Q_{\pi_{\text{ref}}} \right) \right) - \log \pi^{m(y^{<t})} - \frac{1}{\beta} V_{\pi_{\text{ref}}} + \log Z(x, y^{<t})$$

$$= \max_\pi \beta\, \mathbb{E}_{x,y^{<t}\sim D, y^t \sim \pi} \frac{m(y^{<t})}{m(y^{<t})} \log \left( \frac{1}{Z(x,y^{<t})} \pi_{\text{ref}}^{m(y^{<t})} \exp\left( \frac{1}{\beta} Q_{\pi_{\text{ref}}} \right) \right) - m(y^{<t}) \log \pi - \frac{1}{\beta} V_{\pi_{\text{ref}}} + \log Z(x, y^{<t})$$

$$= \max_\pi \beta\, \mathbb{E}_{x,y^{<t}\sim D, y^t \sim \pi} m(y^{<t}) \log \left( \frac{1}{Z(x,y^{<t})} \pi_{\text{ref}}^{m(y^{<t})} \exp\left( \frac{1}{\beta} Q_{\pi_{\text{ref}}} \right) \right)^{\frac{1}{m(y^{<t})}} - m(y^{<t}) \log \pi - \frac{1}{\beta} V_{\pi_{\text{ref}}} + \log Z(x, y^{<t})$$

$$= \max_\pi \beta\, \mathbb{E}_{x,y^{<t}\sim D, y^t \sim \pi} m(y^{<t}) \left( \log \left( \frac{1}{Z(x,y^{<t})} \pi_{\text{ref}}^{m(y^{<t})} \exp\left( \frac{1}{\beta} Q_{\pi_{\text{ref}}} \right) \right)^{\frac{1}{m(y^{<t})}} - \log \pi \right) - \frac{1}{\beta} V_{\pi_{\text{ref}}} + \log Z(x, y^{<t})$$

$$= \max_\pi \beta\, \mathbb{E}_{x,y^{<t}\sim D, y^t \sim \pi} m(y^{<t}) \left( \log \left( \frac{1}{Z(x,y^{<t})} \pi_{\text{ref}} \exp\left( \frac{1}{\beta\, m(y^{<t})} Q_{\pi_{\text{ref}}} \right) \right) - \log \pi \right) - \frac{1}{\beta} V_{\pi_{\text{ref}}} + \log Z(x, y^{<t})$$

$$= \max_\pi \beta\, \mathbb{E}_{x,y^{<t}\sim D, y^t \sim \pi} m(y^{<t}) \log \left( \frac{\frac{1}{Z(x,y^{<t})} \pi_{\text{ref}} \exp\left( \frac{1}{\beta\, m(y^{<t})} Q_{\pi_{\text{ref}}} \right)}{\pi} \right) - \frac{1}{\beta} V_{\pi_{\text{ref}}} + \log Z(x, y^{<t})$$

$$= \max_\pi -\beta\, D_{\text{KL}} \left( \pi \| \frac{1}{Z(x,y^{<t})} \pi_{\text{ref}} \exp\left( \frac{1}{\beta\, m(y^{<t})} Q_{\pi_{\text{ref}}} \right) \right) - \frac{1}{\beta} V_{\pi_{\text{ref}}} + \log Z(x, y^{<t})$$

$$= \min_\pi \beta\, D_{\text{KL}} \left( \pi \| \frac{1}{Z(x,y^{<t})} \pi_{\text{ref}} \exp\left( \frac{1}{\beta\, m(y^{<t})} Q_{\pi_{\text{ref}}} \right) \right) + \frac{1}{\beta} V_{\pi_{\text{ref}}} - \log Z(x, y^{<t}) \tag{9}$$

Where the partition function is given by:

$$Z(x, y^{<t}) = \mathbb{E}_{y^t \sim \pi_{\text{ref}}} \pi_{\text{ref}}(y^t|x, y^{<t})\, \exp\left( \frac{1}{\beta\, m(y^{<t})} Q_{\pi_{\text{ref}}}(y^t|x, y^{<t}) \right)$$

$$= \sum_{y^{<t}} \pi_{\text{ref}}(y^t|x, y^{<t})\, \exp\left( \frac{1}{\beta\, m(y^{<t})} Q_{\pi_{\text{ref}}}(y^t|x, y^{<t}) \right). \tag{10}$$

The objective in Equation 9 can be minimized if the KL term becomes zero (as $Z$ and $V_{\pi_{\text{ref}}}$ are not dependent on $\pi$), which effectively equals to the optimal policy becoming

$$\pi^*(y^t|x, y^{<t}) = \frac{1}{Z(x, y^{<t})} \pi_{\text{ref}}(y^t|x, y^{<t}) \exp\left( \frac{1}{\beta\, m(y^{<t})} Q_{\pi_{\text{ref}}}(y^t|x, y^{<t}) \right). \tag{11}$$

## A.2  DERIVING THE SPARSEPO OBJECTIVE FROM THE BRADLEY-TERRY EQUIVALENCE

The equivalence of Bradley-Terry with the Regret Preference Model, its equivalent on the token-level, has been previously proven in Zeng et al. (2024) as the probability of preferring a chosen response $y_c$ over a rejected response $y_r$,

$$P_{\text{BT}}(y_c > y_r|x) = \sigma\left(\sum_{t=1}^{T_1} A_\pi(y_c^t|x, y_c^{<t}) - \sum_{t=1}^{T_2} A_\pi(y_r^t|x, y_r^{<t})\right) \tag{12}$$

Replacing $A_{\pi_{\text{ref}}}(y^t|x, y^{<t}) \equiv Q_{\pi_{\text{ref}}}(y^t|x, y^{<t}) - V_{\pi_{\text{ref}}}(x, y^{<t})$ in Equation 12 and considering that $V_{\pi_{\text{ref}}}(x, y^{<t}) = \mathbb{E}_{\pi_{\text{ref}}}[Q_{\pi_{\text{ref}}}(y^t|x, y^{<t})]$ we have

$$\sum_{t=1}^{T} A_{\pi_{\text{ref}}}(y^t|x, y^{<t})$$

$$= \sum_{t=1}^{T} Q_{\pi_{\text{ref}}}(y^t|x, y^{<t}) - V_{\pi_{\text{ref}}}(x, y^{<t})$$

$$= \sum_{t=1}^{T} Q_{\pi_{\text{ref}}}(y^t|x, y^{<t}) - \mathbb{E}_{y^t \sim \pi_{\text{ref}}}[Q_{\pi_{\text{ref}}}(y^t|x, y^{<t})] \tag{13}$$

Adding logarithms in front of each part of Equation 11 and solving for $Q_{\pi_{\text{ref}}}$, we get

$$\log \pi^*(y^t|x, y^{<t}) = \log\left(\frac{1}{Z(x, y^{<t})}\pi_{\text{ref}}(y^t|x, y^{<t})\exp\left(\frac{1}{\beta\, m(y^{<t})}Q_{\pi_{\text{ref}}}(y^t|x, y^{<t})\right)\right)$$

$$\log \pi^*(y^t|x, y^{<t}) = \log\left(\frac{1}{Z(x, y^{<t})}\right) + \log \pi_{\text{ref}}(y^t|x, y^{<t}) + \frac{1}{\beta\, m(y^{<t})}Q_{\pi_{\text{ref}}}(y^t|x, y^{<t})$$

$$\log \pi^*(y^t|x, y^{<t}) - \log \pi_{\text{ref}}(y^t|x, y^{<t}) = -\log Z(x, y^{<t}) + \frac{1}{\beta\, m(y^{<t})}Q_{\pi_{\text{ref}}}(y^t|x, y^{<t})$$

$$Q_{\pi_{\text{ref}}}(y^t|x, y^{<t}) = \beta\, m(y^{<t})\log\frac{\pi^*(y^t|x, y^{<t})}{\pi_{\text{ref}}(y^t|x, y^{<t})} + \beta\, m(y^{<t})\log Z(x, y^{<t}) \tag{14}$$

Now, leveraging Equation 14, Equation 13 becomes

$$\sum_{t=1}^{T} A_{\pi_{\text{ref}}}(y^t|x, y^{<t})$$

$$= \sum_{t=1}^{T} \beta\, m(y^{<t})\log\frac{\pi^*(y^t|x, y^{<t})}{\pi_{\text{ref}}(y^t|x, y^{<t})} + \beta\, m(y^{<t})\log Z(x, y^{<t})$$

$$- \mathbb{E}_{y^t \sim \pi_{\text{ref}}}[\beta\, m(y^{<t})\log\frac{\pi^*(y^t|x, y^{<t})}{\pi_{\text{ref}}(y^t|x, y^{<t})} + \beta\, m(y^{<t})\log Z(x, y^{<t})]$$

$$= \sum_{t=1}^{T} \beta\, m(y^{<t})\log\frac{\pi^*(y^t|x, y^{<t})}{\pi_{\text{ref}}(y^t|x, y^{<t})} + \beta\, m(y^{<t})\log Z(x, y^{<t})$$

$$- \mathbb{E}_{y^t \sim \pi_{\text{ref}}}[\beta\, m(y^{<t})\log\frac{\pi^*(y^t|x, y^{<t})}{\pi_{\text{ref}}(y^t|x, y^{<t})}] - \mathbb{E}_{y^t \sim \pi_{\text{ref}}}[\beta\, m(y^{<t})\log Z(x, y^{<t})] \tag{15}$$

Since $m(y^{<t})$ depends only on the previously seen tokens (and not the current one), we can say that $\mathbb{E}_{y^t \sim \pi_{\text{ref}}}[\beta\, m(y^{<t})\, \log Z(x, y^{<t})] = \beta\, m(y^{<t})\, \mathbb{E}_{y^t \sim \pi_{\text{ref}}}[\log Z(x, y^{<t})] = \beta\, m(y^{<t})\, \log Z(x, y^{<t})$. Replacing the above to Equation 15,

$$\sum_{t=1}^{T} A_{\pi_{\text{ref}}}(y^t|x, y^{<t})$$

$$= \sum_{t=1}^{T} \left( \beta\, m(y^{<t}) \log \frac{\pi^*(y^t|x, y^{<t})}{\pi_{\text{ref}}(y^t|x, y^{<t})} - \mathbb{E}_{y^t \sim \pi_{\text{ref}}} \left[ \beta\, m(y^{<t}) \log \frac{\pi^*(y^t|x, y^{<t})}{\pi_{\text{ref}}(y^t|x, y^{<t})} \right] \right)$$

$$= \sum_{t=1}^{T} \left( \beta\, m(y^{<t}) \log \frac{\pi^*(y^t|x, y^{<t})}{\pi_{\text{ref}}(y^t|x, y^{<t})} - \beta\, m(y^{<t})\, D_{\text{KL}}[\pi^*(\cdot|x, y^{<t})\|\pi_{\text{ref}}(\cdot|x, y^{<t})] \right)$$

$$= \sum_{t=1}^{T} \beta\, m(y^{<t}) \log \frac{\pi^*(y^t|x, y^{<t})}{\pi_{\text{ref}}(y^t|x, y^{<t})} - \sum_{t=1}^{T} \beta\, m(y^{<t})\, D_{\text{KL}}[\pi^*(\cdot|x, y^{<t})\|\pi_{\text{ref}}(\cdot|x, y^{<t})]$$

$$= \beta \sum_{t=1}^{T} m(y^{<t}) \log \frac{\pi^*(y^t|x, y^{<t})}{\pi_{\text{ref}}(y^t|x, y^{<t})} - \beta \sum_{t=1}^{T} m(y^{<t})\, D_{\text{KL}}[\pi^*(\cdot|x, y^{<t})\|\pi_{\text{ref}}(\cdot|x, y^{<t})] \tag{16}$$

Finally, replacing the result of Equation 16 that into Equation 12

$$P_{\text{BT}}(y_c > y_r|x) =$$

$$\sigma\Big( \beta \sum_{t=1}^{T_1} m(y_c^{<t}) \log \frac{\pi^*(y_c^t|x, y_c^{<t})}{\pi_{\text{ref}}(y_c^t|x, y_c^{<t})} - \beta \sum_{t=1}^{T_1} m(y_c^{<t})\, D_{\text{KL}}[\pi^*(\cdot|x, y_c^{<t})\|\pi_{\text{ref}}(\cdot|x, y_c^{<t}))$$

$$- \beta \sum_{t=1}^{T_2} m(y_r^{<t}) \log \frac{\pi^*(y_r^t|x, y_r^{<t})}{\pi_{\text{ref}}(y_r^t|x, y_r^{<t})} + \beta \sum_{t=1}^{T_2} m(y_r^{<t})\, D_{\text{KL}}[\pi^*(\cdot|x, y_r^{<t})\|\pi_{\text{ref}}(\cdot|x, y_r^{<t})]\Big) \tag{17}$$

Where we define,

$$u(x, y_c, y_r) = \beta \sum_{t=1}^{T_1} m_u(y_c^{<t}) \log \frac{\pi^*(y_c^t|x, y_c^{<t})}{\pi_{\text{ref}}(y_c^t|x, y_c^{<t})} - \beta \sum_{t=1}^{T_2} m_u(y_r^{<t}) \log \frac{\pi^*(y_r^t|x, y_r^{<t})}{\pi_{\text{ref}}(y_r^t|x, y_r^{<t})} \tag{18}$$

$$\delta(x, y_c, y_r) = \beta \sum_{t=1}^{T_1} m_d(y_c^{<t})\, D_{\text{KL}}[\pi^*(\cdot|x, y_c^{<t})\|\pi_{\text{ref}}(\cdot|x, y_c^{<t})] \tag{19}$$

$$- \beta \sum_{t=1}^{T_2} m_d(y_r^{<t})\, D_{\text{KL}}[\pi^*(\cdot|x, y_r^{<t})\|\pi_{\text{ref}}(\cdot|x, y_r^{<t})]$$

Resulting in

$$p_{BT}(y_c > y_r|x) = \sigma\left(u(x, y_c, y_r) - \delta(x, y_c, y_r)\right) \tag{20}$$

Formulating the maximum likelihood objective given the probability of human preference data in terms of optimal policy in Equation 20, the loss function becomes

$$\mathcal{L}_{SparsePO} = -\mathbb{E}_{(x, y_c, y_r) \sim D}\left[\log \sigma\left(u(x, y_c, y_r) - \delta(x, y_c, y_r)\right)\right] \tag{21}$$

## B  Details on Experimental Setup

In this appendix, we provide further details on the experimental setup. All experiments used AdamW optimizer (Kingma & Ba, 2015).

### B.1  Release

Code is available on: `some.url`.

### B.2  Sentiment Control

**Dataset.** We use the IMDB dataset preprocessed for preference optimization by Amini et al. (2024), which uses prefixes of length 5-8 tokens as prompts.

**Training and Optimization.** All models are trained over three epochs with an effective batch size of 64. For TDPO, we set $\alpha = 0.7$, as it was reported as best for IMDB in Zeng et al. (2024). For DPOP, we set $\lambda = 50$ as reported by Pal et al. (2024).

### B.3  Summary Quality Control

**Dataset.** For preference optimization, we use the TL;DR feedback dataset collected by Stiennon et al. (2020), comprising of two subsets, one with pairwise comparison and the other with the individually rated summaries. Following Amini et al. (2024), we binarize the single-summary subset by selecting the summary with highest and lowest overall Likert score as the chosen and rejected response,respectively. In order to mitigate the compounding effect of summary length, we filtered out training instances with chosen and rejected responses with a length difference greater than 100 words. From these resulting filtered dataset, We uniformly sample 20k and 4k preference instances from each subset to form a training and test set of 40k and 8k instances, respectively.

**Training and Optimization.** All models are trained using LoRA with parameters rank $r = 16$, $\alpha = 16$, and dropout 0.05. Training is done for three epochs with an effective batch size of 256 and learning rate of $1e^{-4}$. We set $\beta = 0.8$ for all systems; $\alpha = 0.5$ for TDPO v1 and v2; weight decay of 0.01 over mask weights; and L1 regularization of 0.001 over all mask values for SparsePO.

**Evaluation.** Statistical significance at the system level is tested pairwise using Bootstrap resampling (Davison & Hinkley, 1997) with a $95\%$ confidence interval. We filter the test set following the methodology in Aharoni et al. (2023) and keep instances with a reference summary–document entailment probability higher than 0.6, given by SummaC$_{ZS}$ Laban et al. (2022).[10] For ROUGE, we report results using stemming; for BERTScore, we use RoBERTa large (Liu et al., 2019) as underlying model with sentence-level IDF importance weighting, for which the scores were calculated over the training set. EDNA scores we calculated using the SummaC$_{ZS}$ score.

Regarding win-rate calculation, we uniformly sample 100 prompts from the entire test set and sample 5 completions using nucleus sampling ($p = 0.95$) and temperatures $T = \{0, 0.25, 0.50, 0.75, 1.0\}$. Then, we elicit quality judgements from GPT4 (gpt-4-turbo) using the prompt in Figure 8, comparing reference summaries against system responses. The order of responses is randomly chosen for each instance.

### B.4  Helpfulness & Harmlessness Control

**Dataset.** We use the Anthropic HH dataset available in HuggingFace.[11]

**Training and Optimization.** The reference model is trained for one epoch over chosen responses with a learning rate of $1e^{-5}$ and an effective batch size of 1024. Preference policy models are trained for three epochs at full precision with an effective batch size of 128, learning rate of $1e^{-6}$, and, otherwise specified, $\beta = 0.1$. For TDPO v1 and v2, we set $\alpha = 0.5$ as it performed better in preliminary experiments. Similarly, we set $\beta = 2.5$ and $\gamma = 0.3$ for SimPO. For SparsePO, we set

---

[10]`https://github.com/tingofurro/summac`
[11]`https://huggingface.co/datasets/Anthropic/hh-rlhf`

```
Which of the following summaries does a better job of summarizing
the most important points in the given forum post, without including
unimportant or irrelevant details?  A good summary is both concise
and precise.

Post:
<post>
Summary A:

<summary_a>

Summary B:
<summary_b>

FIRST provide a one-sentence comparison of the two summaries,
explaining which you prefer and why.  SECOND, on a new line, state
only "A" or "B" to indicate your choice.  Your response should use
the format:
Comparison:  <one-sentence comparison and explanation>
Preferred:  <"A" or "B">
```

Figure 8: Prompt given to GPT4 for win-rate calculation over TL;DR summaries in the test set.

```
For the following query to a chatbot, which response is more
helpful?

Query:  <the user query>

Response A:
<either a system completion or baseline>

Response B:
<the other response>

FIRST provide a one-sentence comparison of the two responses and
explain  which you feel is more helpful.  SECOND, on a new line,
state only "A" or  "B" to indicate which response is more helpful.
Your response should use  the format:
Comparison:  <one-sentence comparison and explanation>
More helpful:  <"A" or "B">
```

Figure 9: Prompt given to GPT4 for win-rate calculation over single-turn dialogue completions in the HH test set.

a learning rate of $5e^{-7}$, mask weight decay of 0.01, and L1 normalization parameter of 0.001 for both reward and KL masks.

**Evaluation.** For OpenLLM leaderboard evaluation, we employ EleutherAI Evaluation Harness library (Gao et al., 2024) and report scores normalized across tasks, as recommended by the leaderboard authors.[12] In this way, individual task scores are reported in the same 0-100 scale, and final average scores are not biased toward one single task.

Similarly to the previous section, we calculate win rates using 100 prompts from the single-turn subset of the test set, sample 5 completions with nucleus sampling ($p = 0.95$) and temperatures $T = \{0, 0.25, 0.50, 0.75, 1.0\}$. Figure 9 shows the prompt used to obtain judgements from GPT4 (gpt-4-turbo), comparing system completions against chosen responses. The order of responses is randomly chosen for each instance.

---

[12]https://huggingface.co/docs/leaderboards/open_llm_leaderboard/normalization

### B.5 Text-to-Code Generation

**Dataset.** The MBPP dataset (Austin et al., 2021)[13] is employed, which consists of 384 train, 90 validation and 500 test programs. We preserve the test set for final evaluation and use the remaining sets for PO training.

**Training and Optimization.** We train StarCoder-1B (Li et al., 2023)[14] for 30 epochs with a learning rate of $5e^{-7}$, a warmup of $10\%$ of the total training steps, linear learning rate decay and an effective batch size of 32.

**Evaluation.** For evaluation we employ the BigCode-evaluation-harness framework (Ben Allal et al., 2022) sampling 100 solutions with temperature $0.6$ and $p = 0.95$. The reported numbers on HumanEval and MBPP are obtained after tuning the $\beta$ values for each method on the $[0.1, 0.2, 0.4, 0.6, 0.8, 1.0, 5.0, 10.0]$ set. The best $\beta$ is obtained based on the performance of each model on pass@10 with 10 samples on HumanEval.

## C Complementary Results

In this appendix, we provide results complementary to our experiments in Section 3.

### C.1 Reward and Response-level KL Divergence Trade-off

In this section, we present further evidence that SparsePO is able to generate responses with higher ground truth reward whilst allowing for larger values of KL divergence, compared to strong PO baselines. Figure 10 presents the case for the sentiment control scenario, showing the relationship between ground truth reward (as given by a sentiment classifier) and response-level KL divergence (i.e., an aggregate of sequence tokens). The plot groups instances in the test set of IMDB by KL divergence level, reporting the average reward per bin, for each system. We compare SparsePO and MaPO against baselines for $\beta = \{0.1, 0.8\}$ and report the following insights. First, at $\beta = 0.1$, DPO exhibits a heavy trade-off between reward for KL divergence, whilst SparsePO$[m_u = m_d]$ and MaPO show similar trade-off to TDPO-v1. Notably, SparsePO$[m_u \neq m_d]$ responses maintain a high level of reward regardless of their KL divergence level. Second, at $\beta = 0.8$, we observe that all DPO and TDPO responses show a KL divergence lower than 10 and a reward of $0.70$. Intriguingly, MaPO does show a heavy reward-KL trade-off, whilst responses generated by SparsePO systems and SimPO maintain high reward levels across all KL levels. The effectiveness of the latter might be explained by the additional $\gamma$ term by which response probabilities are augmented, possibly forcing them to get high enough values that translates to high KL divergence.

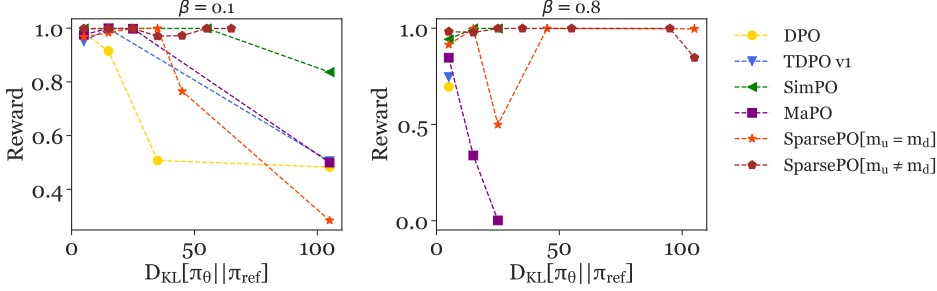

Figure 10: Ground-truth reward of responses grouped by KL divergence range, for responses to the test set of IMDB, for PO systems at $\beta = 0.1$ (left) and $0.8$ (right).

---

[13] https://huggingface.co/datasets/google-research-datasets/mbpp
[14] https://huggingface.co/loubnabnl/starcoder-1b

## C.2   SPARSITY AND TOKEN-LEVEL KL DIVERGENCE

We also report the sparsity levels in the reward and divergence masks, for increasing values of $\beta$, over the *rejected* responses during training for sentiment control in Figure 11.

Figure 12 shows sparsity and token-level KL divergence for chosen responses and Figure 13 for the rejected ones in the code domain. Higher values of $\beta$ do offer significant KL control, resulting into lower KL. Sparsity is much lower for reward masks and higher for KL masks, with both being relatively stable within a small range of values ($\pm$ 4-6 points).

Complementing the discussion in Section 3.2 we can add that, in practice, $\beta$ is acting as the maximum weight we assign to KL restriction, and the mask adjusts it appropriately to each token. We would argue that the mask works in tandem with beta and we observed that the range of betas that are effective with SparsePO is generally higher than DPO (with best values between $0.4-1$). Removing beta ($\beta = 1.0$) results in slightly suboptimal performance.

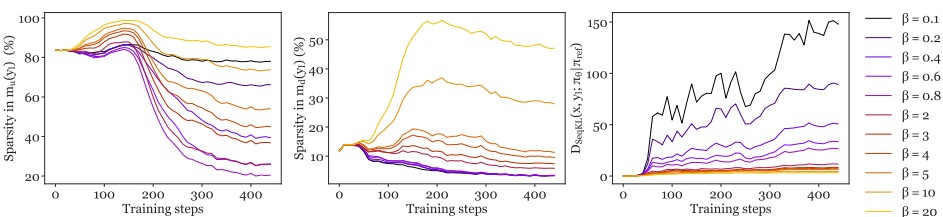

Figure 11: Sparsity levels in the reward mask ($m_u$, left) and the token-level KL divergence mask ($m_d$, middle), as well as token-level KL divergence of *rejected* responses during training (over IMDB), for increasing values of $\beta$.

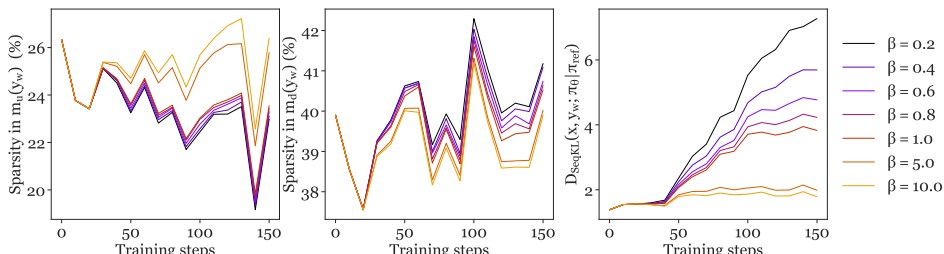

Figure 12: Sparsity levels in the reward mask ($m_u$, left), the token-level KL divergence mask ($m_d$, middle), and token-level divergence of *chosen* responses during training MBPP), for increasing $\beta$.

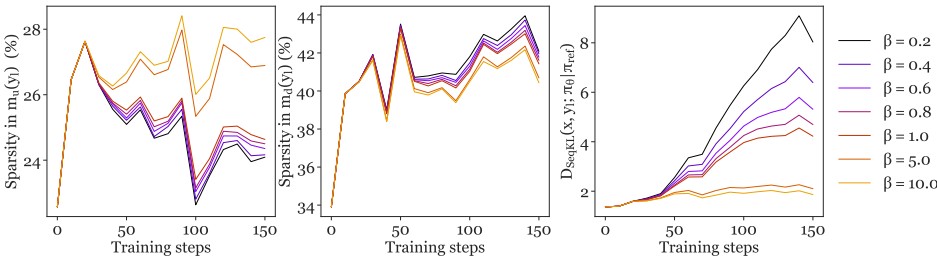

Figure 13: Sparsity levels in the reward mask ($m_u$, left) and the token-level KL divergence mask ($m_d$, middle), as well as token-level KL divergence of *rejected* responses during training (over MBPP), for increasing values of $\beta$.

### C.3 MASK DISTRIBUTION AND TOKEN-LEVEL KL DIVERGENCE

Next, we extend the analyses presented in §3.2, §3.5, and C.2, to investigate the distribution of mask values and token-level KL divergence, for the case of controlled summarization, dialogue, and text-to-code generation. For each task, we report the distribution of mask values over chosen and rejected responses of the corresponding test set, obtained by SparsePO$[m_u \neq m_d]$, SparsePO$[m_u = m_d]$, and MaPO. Additionally, we report the token-level KL divergence during training, as well as the divergence margin, defined as $|D_{SeqKL}(x, y_w; \pi_\theta | \pi_{ref}) - D_{SeqKL}(x, y_l; \pi_\theta | \pi_{ref})|$.

**Controlled Summarization.** Figure 14 shows the mask distributions and Figure 15, the token-level KL divergence for the summarization case. When learned independently (SparsePO$[m_u \neq m_d]$), reward ($m_u$) and KL masks ($m_d$) obtain value distributions with significantly different concentration regions, as shown in Figure 14. The reward mask concentrates its values around 1.0, signifying that for summarization, most response tokens do contribute to the reward. In contrast, the KL mask concentrates in the lower half of its range, indicating that KL is controlled more strictly for most tokens in a response. However, as seen in Figure 15, SparsePO$[m_u \neq m_d]$ obtains higher KL than SparsePO$[m_u = m_d]$ throughout training, possibly indicating that the tokens that SparsePO$[m_u \neq m_d]$ assigned high mask values to were also allowed to diverge more compared to SparsePO$[m_u = m_d]$. Lastly, MaPO showcases a seemingly normal distribution centered on 0.5. This is to be expected since its mask values are derived from the reference model activations.

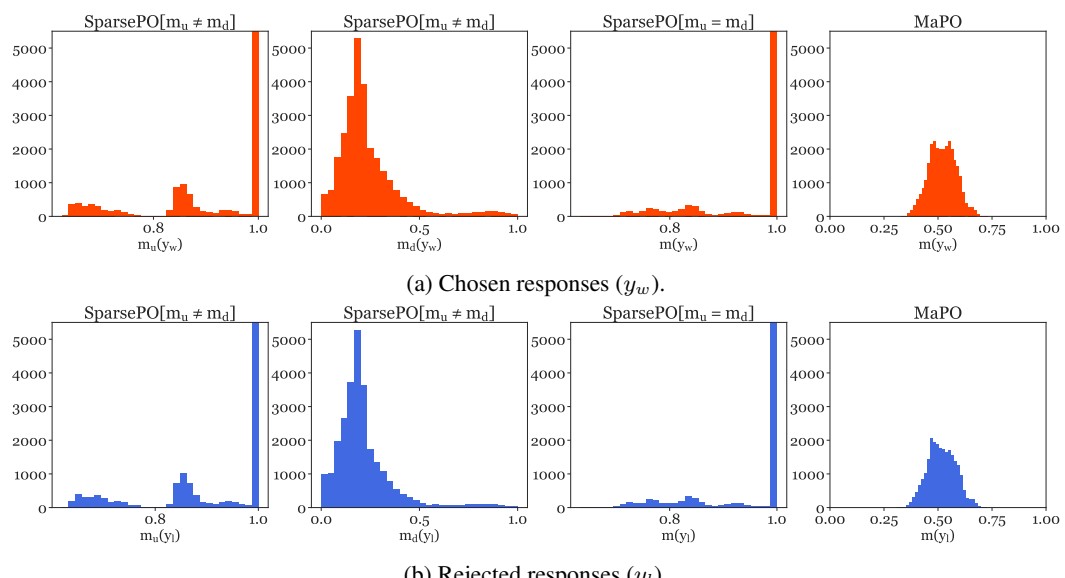

(a) Chosen responses ($y_w$).

(b) Rejected responses ($y_l$).

Figure 14: Distribution of mask values obtained for summarization (TL;DR) in chosen (top) and rejected (bottom) responses. From left to right, SparsePO reward ($m_u$) and KL masks ($m_d$) learned independently (SparsePO$[m_u \neq m_d]$); SparsePO common mask (SparsePO$[m_u = m_d]$); and MaPO mask.

**Helpfulness & Harmlessness Control.** Figure 16 and Figure 17 present mask distributions and token-level KL divergence for the HH case, respectively. For SparsePO$[m_u \neq m_d]$, both the reward ($m_u$) and and KL ($m_d$) masks exhibit values close to zero, with $m_u$ showing a slightly larger range. Similarly, SparsePO$[m_u = m_d]$ obtains values of up to 0.5 but still concentrated at zero. Also, note that the token-level divergence of SparsePO$[m_u = m_d]$ is larger than that of SparsePO$[m_u \neq m_d]$ during training. This means that a lower accumulation of mask values around zero (and hence lower sparsity) allows KL to diverge more in SparsePO$[m_u = m_d]$ than in SparsePO$[m_u \neq m_d]$. The divergence in SparsePO$[m_u \neq m_d]$ is nevertheless significant, showing that, similarly to the summarization case, the few tokens that are allowed to diverge are diverging quite largely.

**Text-to-Code Generation.** Lastly, mask distributions and token-level KL divergence for the code executability case are presented in Figure 18 Figure 19, respectively. We find that the interplay between mask distribution and KL divergence is similar to the HH control case. Both masks

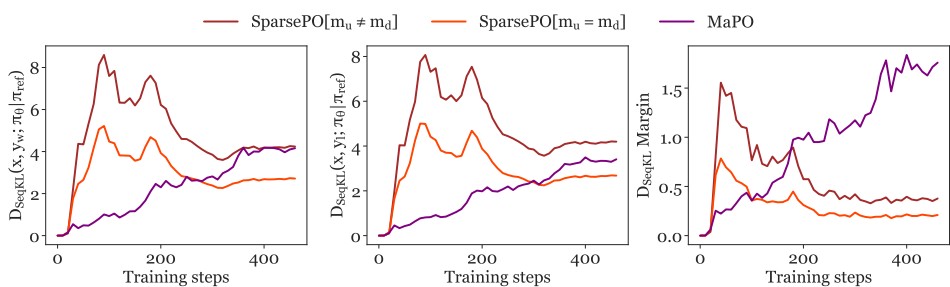

Figure 15: Token-level KL divergence chosen (left) and rejected (middle) responses, as well as the KL margin (right), over TL;DR.

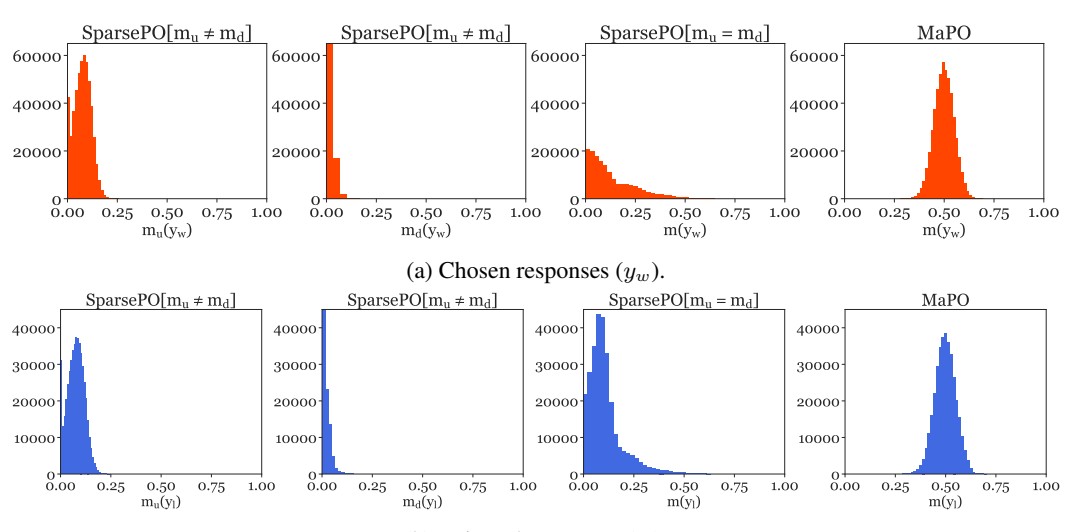

Figure 16: Distribution of mask values obtained for dialogue (Anthropic HH) in chosen (top) and rejected (bottom) responses. From left to right, SparsePO reward ($m_u$) and KL masks ($m_d$) learned independently (SparsePO[$m_u \neq m_d$]); SparsePO common mask (SparsePO[$m_u = m_d$]); and MaPO mask.

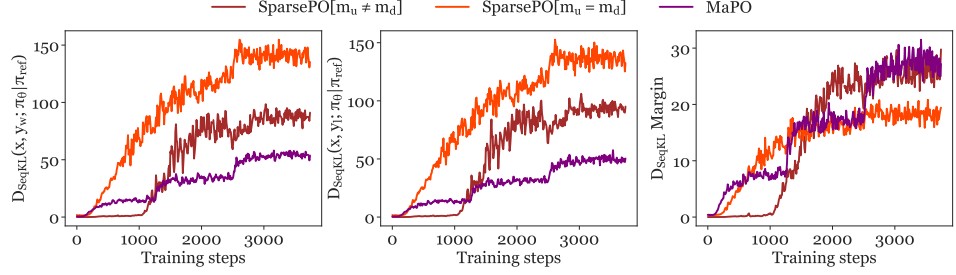

Figure 17: Token-level KL divergence chosen (left) and rejected (middle) responses, as well as the KL margin (right), over Anthropic HH.

in SparsePO[$m_u \neq m_d$] concentrate their values around zero, with $m_u$ showing a wider spread than $m_d$, similar to the behavior of the common mask in SparsePO[$m_u = m_d$]. This means that, when allowed to learn $m_d$ independently from $m_u$, SparsePO implements a stricter control over KL compared to the control over rewards, as also seen in the lower token-level divergence of SparsePO[$m_u \neq m_d$].

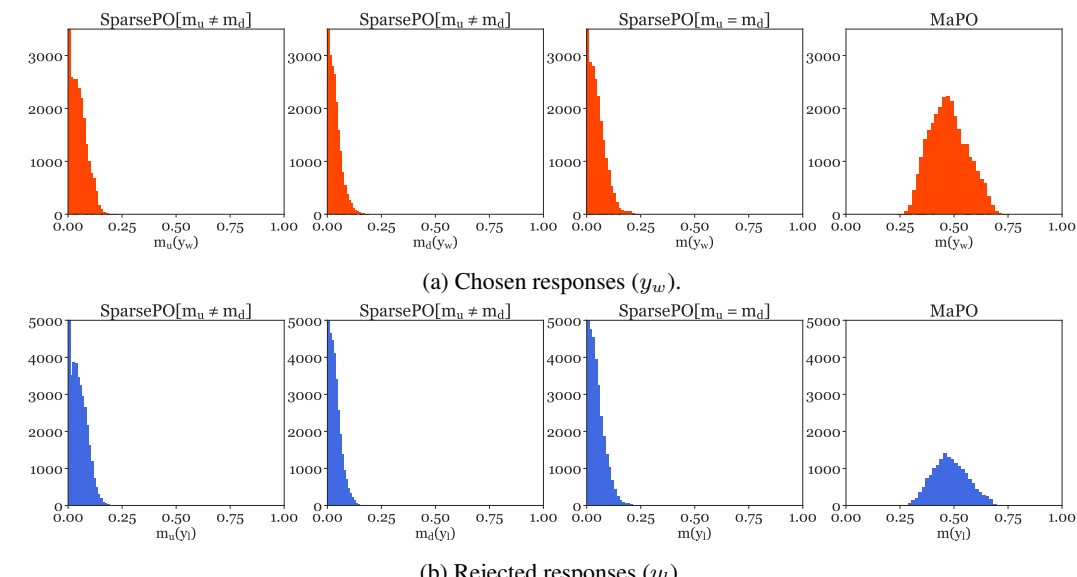

(a) Chosen responses ($y_w$).

(b) Rejected responses ($y_l$).

Figure 18: Distribution of mask values obtained for text-to-code generation (MBPP) in chosen (top) and rejected (bottom) responses. From left to right, SparsePO reward ($m_u$) and KL masks ($m_d$) learned independently (SparsePO[$m_u \neq m_d$]); SparsePO common mask (SparsePO[$m_u = m_d$]); and MaPO mask.

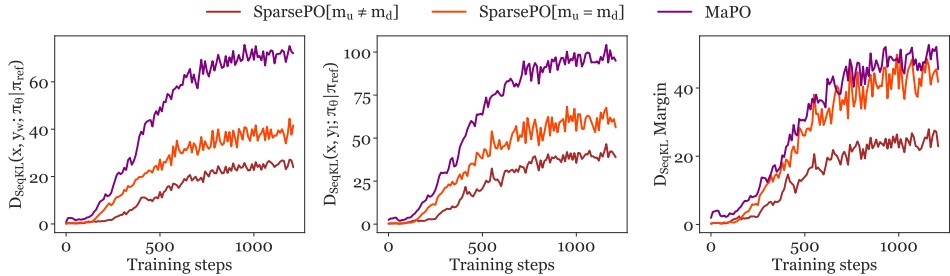

Figure 19: Token-level KL divergence chosen (left) and rejected (middle) responses, as well as the KL margin (right), over MBPP.

### C.4    RESULTS ON OPEN LLM LEADERBOARD V1

Complementary to Open LLM Leaderboard v2, we report results on the original version of the leaderboard, since we primarily experiment with small-sized models (<2B parameters). In Table 3 we observe that all methods obtain improved scores over the SFT baseline (which was not the case in v2), with most notably improvements in Winogrande with SparsePO[$m_u = m_d$].

### C.5    SUMMARY QUALITY CONTROL

We report results complementary to § 3.4, for completeness. Figure 20 shows similar metrics to Figure 6 but over the entire test set of TL;DR. SparsePO and MaPO obtain comparable levels of relevancy and diversity than all other models. Contrary to the more controlled setup in Fig. 6, SparsePO and MaPO do fall behind other models in terms of EDNA scores for low temperatures. Note that these results are obtained over all test instances, regardless of their level of document-reference summary faithfulness.

| METHODS | ARC | HELLASWAG | TRUTHFULQA | MMLU | WINOGRANDE | AVG |
|---|---|---|---|---|---|---|
| SFT | 26.52 | 46.74 | 41.63 | 22.49 | 56.43 | 38.76 |
| DPO | 27.61 | 47.64 | 42.35 | 23.87 | 56.80 | 39.65 |
| TDPO v1 | 30.20 | 49.05 | 41.35 | 24.11 | 56.09 | 40.16 |
| TDPO v2 | 28.95 | 48.61 | 43.14 | 23.48 | 56.27 | 40.09 |
| SIMPO | 28.50 | 33.07 | **47.73** | 23.21 | 51.93 | 36.38 |
| DPOP | **30.38** | 47.91 | 43.48 | 22.83 | 56.09 | 40.13 |
| MAPO | 29.10 | **50.89** | 41.63 | 24.63 | 57.77 | **40.80** |
| SPARSEPO$[m_u = m_d]$ | 28.73 | 48.48 | 42.23 | **24.91** | **59.12** | 40.69 |
| SPARSEPO$[m_u \neq m_d]$ | 29.92 | 47.15 | 42.97 | 23.64 | 57.46 | 40.22 |

Table 3: Performance of Pythia 1.4B models on Open LLM Leaderboard 1 after PO with Helpfulness & Harmlessness as proxy for human preference. Best number across PO methods are bolded.

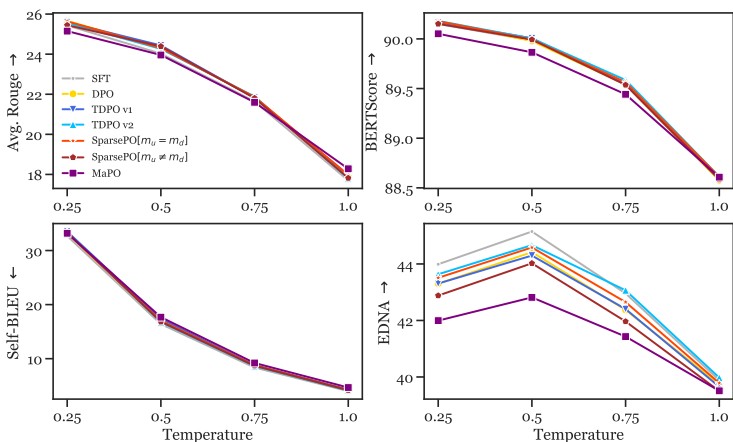

Figure 20: Performance of summarization models in terms of relevance (avg. ROUGE F1, BERTScore), lexical diversity (Self-BLEU), and faithfulness and diversity (EDNA + SummaC$_{ZS}$), across temperature values ($x$ axis), over the complete test set of TL;DR.

## C.6    QUALITATIVE ANALYSIS

Figure 21 presents complementary results to Figure 4b, showcasing mask values per token in rejected response examples, for the case of sentiment control.=

**Mask behavior and response correctness.** Next, we analyze the behavior of the mask in scenarios where the 'correctness' of the task can be verified deterministically, taking as test cases the tasks text-to-code generation and mathematical reasoning. Both of these tasks require that a response is 'correct', however with a crucial difference. In current math benchmarks (e.g. MATH) correctness is evaluated as obtaining the correct final answer, regardless of the correctness of intermediate reasoning steps. Hence, a model has more liberty in generating a response consisting of steps and the final answer, i.e. if a response contains incorrect intermediate steps but the correct final answer, it will be deemed as correct. However, in our text-to-code setup, it is crucial that the response not only executes but also that it returns the correct answer for all test units. In this case, an incorrect intermediate logical step in the response, even if executable, will prompt an incorrect answer (or fail to run).

Based on this intuition, we hypothesize that SparsePO struggles in cases where the response consists of formal language or rigorous steps, i.e. where there is little to no leeway for generation diversity. Figure 22 shows the mask values for responses in HH control, code generation, and algebraic reasoning. The latter example was taken from the MATH dataset Hendrycks et al. (2021) and derived using our Pythia-1.4B model trained over HH. In the first example, showing a response to a query in HH, the mask accentuates relevant tokens in the response (e.g. *consists of*, *vegetables*). In the second example, algebraic reasoning, the mask manages to accentuate relevant operators and intermediate results and, more strongly, natural prose. Finally, the last example shows that programming

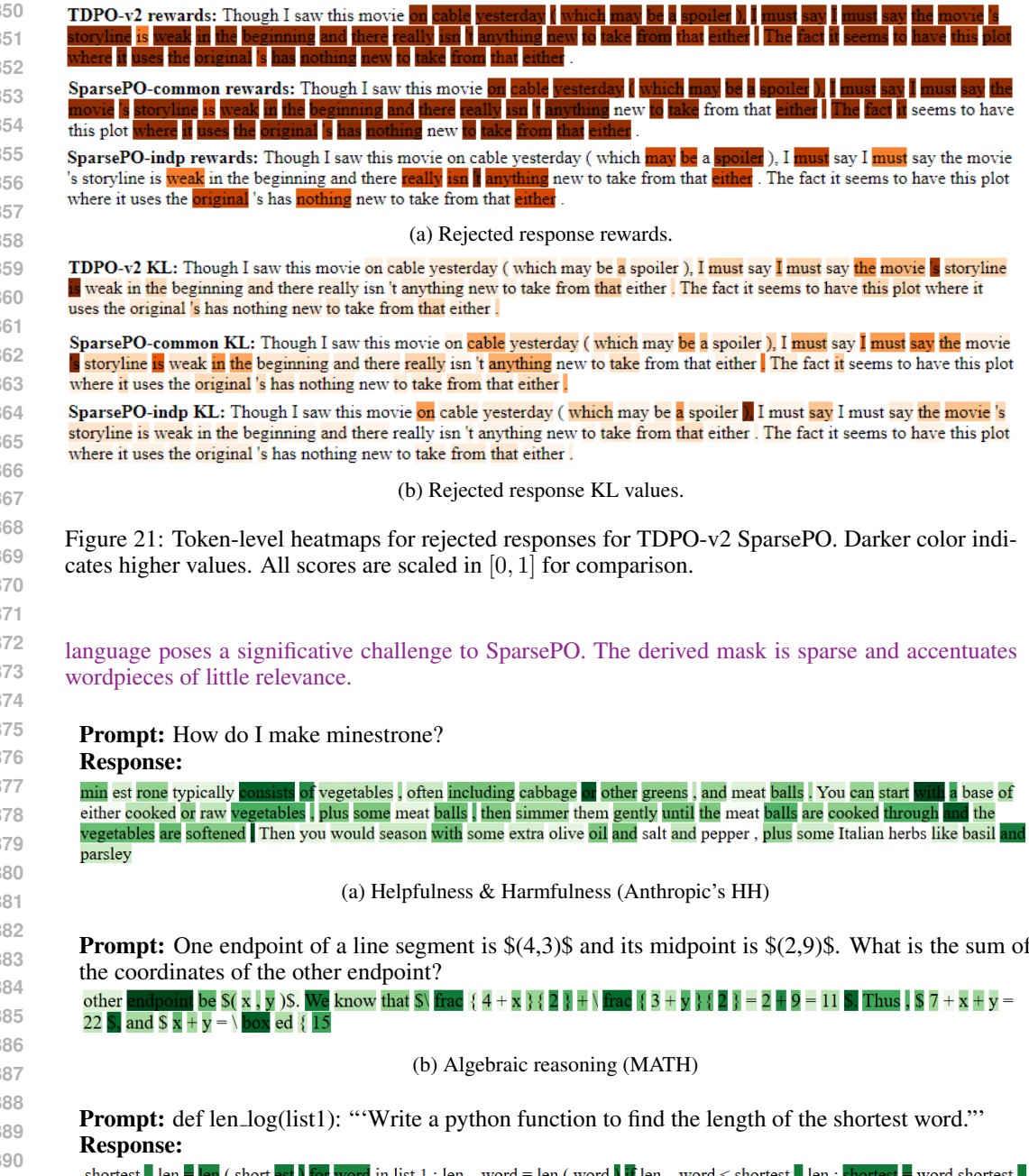

(a) Rejected response rewards.

(b) Rejected response KL values.

Figure 21: Token-level heatmaps for rejected responses for TDPO-v2 SparsePO. Darker color indicates higher values. All scores are scaled in $[0, 1]$ for comparison.

language poses a significative challenge to SparsePO. The derived mask is sparse and accentuates wordpieces of little relevance.

**Prompt:** How do I make minestrone?
**Response:**

min est rone typically consists of vegetables , often including cabbage or other greens , and meat balls . You can start with a base of either cooked or raw vegetables , plus some meat balls , then simmer them gently until the meat balls are cooked through and the vegetables are softened . Then you would season with some extra olive oil and salt and pepper , plus some Italian herbs like basil and parsley

(a) Helpfulness & Harmfulness (Anthropic's HH)

**Prompt:** One endpoint of a line segment is $(4,3)$ and its midpoint is $(2,9)$. What is the sum of the coordinates of the other endpoint?

other endpoint be $(x, y)$. We know that $\frac{4+x}{2} + \frac{3+y}{2} = 2 + 9 = 11$ $ Thus , $7 + x + y = 22$, and $x + y = \boxed{15$

(b) Algebraic reasoning (MATH)

**Prompt:** def len_log(list1): '''Write a python function to find the length of the shortest word.'''
**Response:**

shortest len = len ( short est ) for word in list 1 : len _ word = len ( word ) if len _ word < shortest len : shortest = word shortest len = len word return shortest

(c) Text-to-code generation (MBPP)

Figure 22: Token-level mask values obtained by SPARSEPO$[m_u = m_d]$ over chosen responses in HH, MBPP, and MATH. Darker color indicates higher mask value.

## C.7 HUMANRANKEVAL EVALUATION

We further report results on the HumanRankEval benchmark (Gritta et al., 2024) in Table 4. The reported categories correspond to Unix-based OS (UNIX), English Language (ENG.), Physics, LaTeX, Software Engineering (S.ENG.), Maths and Statistics (STATS), CS+DB (CodeReview, Computer Science, Data Science and Databases), Apple and Android (A+A) and Lang+Sci (Latin, Chinese,

French, German, Japanese, Spanish plus Engineering, Chemistry, Biology, Earth Science and Astronomy).

| METHODS | A+A | C++ | CS+DB | ENG. | HTML | JAVA | LANG+SCI | LATEX | MATH | PHYSICS | PYTHON | S.ENG. | STATS | UNIX | AVG |
|---|---|---|---|---|---|---|---|---|---|---|---|---|---|---|---|
| PYTHIA-1.4B | 10.15 | 14.66 | 8.46 | 12.52 | 11.27 | 10.84 | 12.76 | 16.55 | 13.70 | 12.43 | 9.47 | 9.60 | 13.78 | 11.71 | 11.99 |
| SFT | 10.61 | 14.87 | 8.82 | 12.27 | 12.23 | 11.21 | 13.26 | 16.10 | 13.34 | 12.18 | 9.37 | 9.22 | 13.40 | 11.59 | 12.03 |
| DPO | 11.36 | 15.20 | 10.09 | 11.44 | 13.39 | 11.41 | 13.74 | 16.64 | 13.33 | 12.25 | 9.82 | 9.99 | 14.13 | 11.86 | 12.47 |
| TDPO-v1 | 11.28 | 15.14 | 9.35 | 11.39 | 12.56 | 11.17 | 13.30 | 16.31 | 13.52 | 12.36 | 9.33 | 9.80 | 13.79 | 11.67 | 12.21 |
| TDPO-v2 | 10.64 | 14.88 | 9.09 | 11.85 | 12.59 | 11.12 | 13.25 | 16.15 | 13.58 | 12.12 | 9.07 | 9.30 | 13.77 | 11.60 | 12.07 |
| DPO-P | 11.11 | 15.15 | 9.45 | 11.81 | 12.83 | 11.47 | 13.51 | 16.45 | 13.57 | 12.33 | 9.66 | 9.54 | 14.06 | 11.96 | 12.35 |
| SIMPO | 3.35 | 7.68 | 3.99 | 6.04 | 6.29 | 2.79 | 4.80 | 5.26 | 2.69 | 6.32 | 7.57 | 2.97 | -1.69 | 8.20 | 4.73 |
| MAPO | 11.19 | 15.03 | 10.50 | 10.73 | 13.05 | 11.62 | 13.32 | 16.27 | 13.60 | 12.52 | 9.66 | 10.74 | 13.81 | 11.45 | 12.39 |
| SPARSEPO$[m_u = m_d]$ | 11.23 | 15.45 | 9.80 | 11.37 | 13.38 | 11.55 | 13.73 | 15.80 | 13.23 | 11.72 | 10.12 | 10.35 | 13.84 | 11.25 | 12.34 |
| SPARSEPO$[m_u \neq m_d]$ | 12.94 | 17.09 | 11.27 | 12.52 | 14.68 | 13.99 | 15.08 | 17.52 | 13.86 | 12.34 | 12.48 | 9.58 | 15.39 | 13.19 | 13.71 |

Table 4: Performance of Pythia 1.4B models on HumanRankEval after PO with Helpfulness & Harmlessness as proxy for human preference.

# D ABLATION STUDIES

In this section, we present ablation studies that investigate the contribution of design choices in mask architectures. All experiments were done by performing SFT and PO training on Pythia-410M using the DPO-mix-7k dataset curated by Argilla.[15] This dataset consists of 7k instances mixed from Capybara[16] a synthetic multi-turn dialogue dataset; Intel ORCA[17], a single-turn dataset based on FLAN, with prompts aiming at helpful, truthful, and verbalized calibration; and the binarized, filtered version of UltraFeedback.[18] Training was done for three epochs with learning rate of $5e-7$ and effective batch size of $128$ for all models. Unless otherwise stated, all SparsePO systems were trained using the common mask setup.

## D.1 MASK ARCHITECTURE

We experiment with the number of model layers used for mask calculation, as well as the number of feedforward layers in the mask architecture itself. Table 5 showcases the performance of our design choices over benchmarks in the OpenLLM learderboard v2.

| Lay.per Mask | #FF$_m$ | BBH | MATH | GPQA | MuSR | MLMU pro | IFEval Instr. | IFEval Prom. | Avg. |
|---|---|---|---|---|---|---|---|---|---|
| All Layers | 1 | 4.60 | 0.91 | 1.68 | 12.47 | 1.57 | 21.70 | 11.28 | **7.74** |
| Last Layer | 1 | 4.34 | 0.68 | 2.01 | 11.74 | 1.41 | 19.42 | 9.61 | 7.03 |
| Last Layer | 2 | 4.60 | 0.98 | 1.68 | 11.57 | 1.24 | 19.30 | 9.61 | 7.00 |

Table 5: OpenLLM leaderboard v2 performance of mask architectural choices, for Pythia 410M-based models trained over DPO-mix-7k.

## D.2 HYPER-PARAMETER TUNING

Next, we investigate the effect of weight decay regularization applied over the mask, with results shown in Table 6.

## D.3 BINARY AND RANDOM MASKS

Finally, we experiment with variations of SparsePO in which the learned mask is replaced by a uniformly-sampled random vector with values between $[0,1]$, and a learned binary mask with a $sign$ activation function, i.e. the mask is set to 1 for all positive values and 0, otherwise. Table 7 presents the results over the OpenLLM leaderboard.

---

[15] https://huggingface.co/datasets/argilla/dpo-mix-7k
[16] https://huggingface.co/datasets/argilla/distilabel-capybara-dpo-7k-binarized
[17] https://huggingface.co/datasets/argilla/distilabel-intel-orca-dpo-pairs
[18] https://huggingface.co/datasets/argilla/ultrafeedback-binarized-preferences-cleaned

| Wgt. Decay | BBH | MATH | GPQA | MuSR | MLMU pro | IFEval Instr. | Prom. | Avg. |
|---|---|---|---|---|---|---|---|---|
| 0 | 4.44 | 0.83 | 1.57 | 13.39 | 1.48 | 22.42 | 11.09 | 7.89 |
| 0.001 | 4.41 | 0.38 | 1.45 | 11.47 | 1.61 | 21.82 | 11.09 | 7.46 |
| 0.01 | 4.56 | 0.38 | 1.12 | 14.00 | 1.36 | 23.02 | 12.57 | **8.14** |
| 0.1 | 4.83 | 0.68 | 1.34 | 12.03 | 1.66 | 21.82 | 10.91 | 7.61 |
| 1.0 | 4.65 | 0.68 | 1.45 | 12.70 | 1.64 | 21.70 | 10.72 | 7.65 |

Table 6: OpenLLM leaderboard v2 performance for several levels of weight decay regularization over the mask.

| Mask | BBH | MATH | GPQA | MuSR | MLMU pro | IFEval Instr. | Prom. | Avg. |
|---|---|---|---|---|---|---|---|---|
| SparsePO$[m_u = m_d]$ | 4.56 | 0.38 | 1.12 | 14.00 | 1.36 | 23.02 | 12.57 | **8.14** |
| SparsePO[Binary] | 4.55 | 1.13 | 1.68 | 13.03 | 1.46 | 18.71 | 8.50 | 7.01 |
| Random | 4.84 | 0.68 | 1.34 | 14.49 | 1.33 | 20.26 | 9.61 | 7.51 |

Table 7: OpenLLM leaderboard v2 performance for binary and random masks.

