# OpenReview forum: "SparsePO: Controlling Preference Alignment of LLMs via Sparse Token Masks"
_ICLR.cc/2025/Conference — Submitted to ICLR 2025_

### Official Review · Reviewer_hU2n · 2024-10-29

**Soundness:** 2
**Presentation:** 2
**Contribution:** 1
**Rating:** 3
**Confidence:** 4

**Summary:**

* This paper introduces token-level masking to the token-level DPO objective proposed by Zeng, et. a (2024)l. The motivation is that not all tokens should contribute to KL divergence and Reward computation equally.
* They conduct experiments on standard tasks for preference optimization: Sentiment Control, Helpfulness & Harmlessness Control, TL;DR summarization, and Text-to-Code generation.
* They analyze Sparisity level (i.e,. the amount of zero maskings) of the learned masks during training and discuss its implication.
* Their experimental results suggest that DPO remains a strong baseline. The proposed methods are marginally better, if at all than DPO, in terms of final performance or KL-Performance frontier,

**Strengths:**

* They select a broad set of experiments for investigating PO methods.
* The proposed token-level masking method is a natural extension of previous work on token-leval DPO.

**Weaknesses:**

* **Lack of precision in discussing results**. The discussion heavily relies on imprecise statements such as “TDPOv1 exhibits moderate KL divergence”. It is not clear how an PO algorithm could exhibit a certain level of KL divergence. This results in serious issues in clarity as to what the paper tries to argue with its experimental results (See Questions Below).
* **Proposed methods show marginal or no performance gain compared to DPO**. There is little evidence for adopting the proposed methods in terms of performance. For example, in sentiment control, DPO shows the best KL-Reward frontier at low KL (<10). Experiments on Text-to-Code (Table 2) essentially shows DPO remains the most robust PO algorithm. It is difficult to justify the complicated modeling techniques introduced by token-level masking based on their experimental results.
* **Inadequate experimental setups**. While the paper includes a good range of tasks, some experimental setups are inadequate for investigating PO. For example, Table 1 on Helpfulness & Harmlessness show all PO methods underperforming SFT policy by substantial margin in terms of average scores on Open LLM Leaderboard 2. It is inadequate to argue which PO algorithm is best on the ground that it induces less degradation than SFT compared to other methods. In addition, Helpfulness & Harmfulness and Text-to-Code are investigated with models with <2B parameters. The baseline performance of these small models is poor on these tasks.
* **Advantages from introducing token-level masking are unclear**.  It is not clear from the paper what theoretical/empirical advantages there are for the proposed token-leval masks.

**Questions:**

> Line 242: “TDPOv1 exhibits moderate KL divergence, which translates into higher reward than DPO and comparable to SimPO.”
1. It looks to me that DPO attains substantially higher reward at KL <10 in Figure 2 than all other PO methods except MaPO. Could you explain?

2.  Could you explain what “TDPOv1 exhibits moderate KL divergence, which translates into higher reward than DPO ...” mean? Do you mean systems trained with TDPOv1 generally have KL < 20 at the end of TDPOv1 training?

3. In stating “moderate KL translate into higher reward”, Are you suggesting a causal relationship between KL and reward? I understand KL and reward are two measures of system characteristics which could be correlated, but without causal relation.

> Line 302: “increasingly higher values of β induce higher levels of sparsity on the divergence mask (md), restricting the amount of tokens allowed to diverge in a sequence, which translates to lower token-level KL divergence throughout training.”

4. From Equation (3): high sparsity (i.e., more zero maskings) effectively drops out the KL term, allowing the policy to optimize the advantage function only. It seems like this increases the amount of tokens allowed to diverage in a sequence rather than restricting it. Could you help me understand how mask sparsity interferes with the policy objective?

> Line 307: “we find that low values of β induce scenarios where reward sparsity is high and divergence sparsity is low, meaning that the loss is dominated by the masked divergence term, δ(x,y1,y2).”

5. Do you mean that the mask is restricting KL divergence at small beta? It seems like the mask is working against beta’s control of regularization.
6. I am not sure what are the theoretical and emprical advantages of the proposed SPARSEPO methods. Could you provide a clear summary?

---

> ### Author Response · Authors · 2024-11-18
> **Author Response**
>
> Thank you for taking the time to review our paper and for the insightful comments.
>
> - [W1] Thank you for this remark. We believe this is easily addressed by us going through the paper, and
> changing some expressions to be more precise. In this particular example you raise, we are referring to
> “moderate KL divergence” in comparison to the other PO methods we examine and not on its own. All
> our KL comparisons are relative and we do not rely on the actual values they obtain.
>
> - [W2] First of all, as it has been shown in Rafailov et al. [2023], the reward-KL frontier should not be necessarily
> used to measure performance of a PO algorithm; it is mostly provided as an analysis of the algorithm’s
> expected behavior. Nevertheless, in Figure 2, the best reward frontier is achieved by MaPO which is our
> dense proposed approach for mask calculation. On the contrary, SparsePO achieves the best reward on
> high KL values compared to other methods. Finally, with respect to code, we acknowledge that in this
> particular domain the method is not performing adequately.
>
> - [W3] To further strengthen the contribution of the paper we have included additional win-rates for HH and
> TL;DR datasets (please look at the general response). We are also in the process of performing evaluation
> on the OLLM-v1 framework, which might be more suitable for $>$2B parameter models. We will update
> our response with the results before the end of the rebuttal period.
>
> - [W4] As mentioned in our responses to Reviewers 5gHZ and 6TmP, previous work in modeling preference at
> finer levels than complete responses has shown benefits in downstream tasks. In this context, our method
> presents the following advantages. First, we introduce an explicit way of controlling for the preference of
> each token during optimization. Intuitively, certain preference proxies (e.g. toxicity, sentiment) can be
> better modeled when the optimization signal is constrained to the most indicative tokens in the response
> instead of the entire response, potentially filtering out noisy signals. In our method, token preference is controlled by a specialized mask, having the option to leverage the model activations (MaPO, non-learned) or learning a sparse mask (SparsePO). When learned, the resulting masks achieve a level of sparsity optimal to a specific preference proxy. Please also refer to the general response to reviewers where we summarize our contributions and provide additional results.
>
> - [Q1] In Figure 2 we report the trade-off between response-level KL divergence and reward aiming to visualize
> their Pareto frontier. In order to obtain this frontier, we use an existing sentiment classifier that acts
> as our ground-truth reward - a process that has been also used in the original DPO paper. Lower KL
> ranges indicate less divergence from the reference model and high rewards indicate that the method
> is able to generate more samples with positive sentiment. We find that MaPO is able to produce the
> most examples with positive sentiment while not diverging from the reference (high reward, low KL).
> On the other hand, SparsePO trades-off reward values for KL - meaning that when the model diverges
> from the reference model, SparsePO is still able to generate more positive reviews compared to other
> methods. This underlines the desired behavior of our method–by allowing more tokens to diverge from
> the reference we enable the policy to generalize to new answers.
> As mentioned in a previous response, this reward and divergence should be used to analyse the behavior
> of the approach, and not necessarily to evaluate its efficacy.
>
> - [Q2] The comparisons we make for KL divergence are relative and not based on hard thresholds. We observe
> that TDPO-v1 overall does achieve lower KL compared to SimPO, DPOP and SparsePO, bringing to a
> middle point in regards with the other reported PO algorithms.
>
> - [Q3] Thank you for pointing this out. This is not phrased in the best possible way, as we did not mean to
> claim any causal relation. We will rephrase to stress the correlation between KL divergence and obtained
> expected reward instead.

---

> > ### Author Response · Authors · 2024-11-18
> > **Author Response pt2**
> >
> > - [Q4] After deriving the objective, the loss function is the one described in Equation 6, that incorporates the
> > mask effectively in front of both the rewards and the KL divergence. By design, our model cannot reach
> > 100% sparsity given that the mask values need to be a number close to zero but not zero (line 120). If
> > zero values were allowed, that would render the objective non-optimizable. To further show the interplay
> > between sparsity and policy objective, please consider the Table below, which shows the progression of
> > loss terms and sparsity levels across training of SparsePO in common mask setup over the IMDB dataset
> > with beta=0.1. Terms U and D are the reward and divergence terms in Equation (6), and ’Sp. $m(y_w)$’
> > denotes the sparsity level (%) in mask $m(y_w)$. We can observe that, as expected, sparsity has a direct
> > impact on terms U and D, with lower sparsity eliciting higher values. Crucially, note that none of U or
> > D collapse, instead increasing proportionally as sparsity decreases. It is also worth noting that the values
> > of U and D are close to zero at the beginning of training, when the learning rate is close to zero, since
> > we follow a triangular scheduling regime.
> >
> > | Step | U | D |Sp. m(y_w) | Sp. m(y_l)) |
> > |------|-------|-------|-------|-------|
> > | 0 | 0 | 0 | 91.99 | 91.68 |
> > | 50 | 0.15 | 0 | 87.59 | 87.78 |
> > | 100 | 0.69 | -2.24 | 64.00 | 64.23 |
> > | 150 | 7.21 | -2.61 | 41.63 | 42.07 |
> > | 200 | 31.38 | 15.59 | 25.25 | 25.02 |
> > | 250 | 33.19 | 13.97 | 17.25 | 17.74 |
> > | 300 | 34.81 | 12.66 | 14.26 | 14.48 |
> > | 350 | 64.94 | 34.75 | 11.69 | 11.26 |
> > | 400 | 63.44 | 32.91 | 10.36 | 10.15 |
> >
> > - [Q5] In practice, beta is acting as the maximum weight we assign to KL restriction, and the mask adjusts it
> > appropriately to each token. We would argue that the mask works in tandem with beta and we observed
> > that the range of betas that are effective with SparsePO is generally higher than DPO (with best values
> > between 0.4-1). Removing beta (β = 1.0) results in slightly suboptimal performance.
> >
> > - [Q6] We kindly ask that you refer to the general response to reviewers for a summary of the method’s
> > advantages.

---

> > > ### Comment · Reviewer_hU2n · 2024-11-23
> > >
> > > > Line 302: “increasingly higher values of β induce higher levels of sparsity on the divergence mask (md), restricting the amount of tokens allowed to diverge in a sequence, which translates to lower token-level KL divergence throughout training.”
> > > >> From Equation (3): high sparsity (i.e., more zero maskings) effectively drops out the KL term, allowing the policy to optimize the advantage function only. It seems like this increases the amount of tokens allowed to diverage in a sequence rather than restricting it. Could you help me understand how mask sparsity interferes with the policy objective?
> > > >>> [Q4] After deriving the objective, the loss function is the one described in Equation 6, that incorporates the mask effectively in front of both the rewards and the KL divergence. By design, our model cannot reach 100% sparsity given that the mask values need to be a number close to zero but not zero (line 120). If zero values were allowed, that would render the objective non-optimizable. To further show the interplay between sparsity and policy objective, please consider the Table below, which shows the progression of loss terms and sparsity levels across training of SparsePO in common mask setup over the IMDB dataset with beta=0.1. Terms U and D are the reward and divergence terms in Equation (6), and ’Sp.
> > > ’ denotes the sparsity level (%) in mask
> > > . We can observe that, **as expected, sparsity has a direct impact on terms U and D, with lower sparsity eliciting higher values.** Crucially, note that none of U or D collapse, instead increasing proportionally as sparsity decreases. It is also worth noting that the values of U and D are close to zero at the beginning of training, when the learning rate is close to zero, since we follow a triangular scheduling regime.
> > >
> > > Thanks for this and so empirical results suggest that lower sparsity leads to higher KL and Rewards. I would have expected based on this analysis, the paper would attempt to find good performing models at lower KL than other PO methods. Because otherwise the masks are not sparse (sparsity% < 20%) as suggested by the provided Table.
> > >
> > > In the general response the authors write:
> > >
> > > > Our method produces a **non-trivial level of sparsity** that confirms, alongside the reported performance gains, that not all tokens are required for effective preference optimization.
> > >
> > > Could you point me to specific systems in your results that have non-trivial level of sparsity and with substantial gains compared to other PO methods? I think these should be highlighted, but are not clear in the current version.
> > >
> > >
> > > > Do you mean that the mask is restricting KL divergence at small beta? It seems like the mask is working against beta’s control of regularization.
> > > >> [Q5] In practice, beta is acting as the maximum weight we assign to KL restriction, and the mask adjusts it appropriately to each token. We would argue that the mask works in tandem with beta and we observed that the range of betas that are effective with SparsePO is generally higher than DPO (with best values between 0.4-1). Removing beta (β = 1.0) results in slightly suboptimal performance.
> > >
> > > Thanks for this. I think including this discussion of how beta and the mask coordinate will be helpful for the reader, as it is not immediately clear how they work together.

---

> > > > ### Comment · Reviewer_hU2n · 2024-11-23
> > > > **Comment on General Response**
> > > >
> > > > With respect to the paper's contribution:
> > > >
> > > > > Our method produces a non-trivial level of sparsity that confirms, alongside the reported performance gains, that not all tokens are required for effective preference optimization.
> > > >
> > > > Please see question above. It would be good to show that. But the current results do not highlight this contribution.
> > > >
> > > > > Our method is able to produce higher rewards at larger KL divergence values enabling models to generate more diverse responses.
> > > >
> > > > Why can't other PO methods such as DPO achieve this by using a smaller beta value? And I am not sure what results in the paper specifically support the claim.
> > > >
> > > > > +10% winrates in Anthropic-HH single-turn dialogue (using GPT-4-turbo as a judge across different temperatures).
> > > >
> > > > What is the reference performance that the +10% is measured? It will be helpful to put together a table showing the best performing model produced by each method and controlled for sampling temperature. From my reading, the claim is based on comparing SparsePO to SFT rather than to DPO.

---

> > ### Comment · Reviewer_hU2n · 2024-11-23
> > **Reviewer's Response**
> >
> > Thanks for your response. To make sure we are following the discussion, I have put together your response and my original comment for each point.
> >
> > > Line 242: “TDPOv1 exhibits moderate KL divergence, which translates into higher reward than DPO and comparable to SimPO.”
> > > 1. It looks to me that DPO attains substantially higher reward at KL <10 in Figure 2 than all other PO methods except MaPO. Could you explain?
> > > 2. Could you explain what “TDPOv1 exhibits moderate KL divergence, which translates into higher reward than DPO ...” mean? Do you mean systems trained with TDPOv1 generally have KL < 20 at the end of TDPOv1 training?
> > > 3. In stating “moderate KL translate into higher reward”, Are you suggesting a causal relationship between KL and reward? I understand KL and reward are two measures of system characteristics which could be correlated, but without causal relation.
> >
> > >> [Q1] In Figure 2 we report the trade-off between response-level KL divergence and reward aiming to visualize their Pareto frontier. In order to obtain this frontier, we use an existing sentiment classifier that acts as our ground-truth reward - a process that has been also used in the original DPO paper. Lower KL ranges indicate less divergence from the reference model and high rewards indicate that the method is able to generate more samples with positive sentiment. We find that MaPO is able to produce the most examples with positive sentiment while not diverging from the reference (high reward, low KL). On the other hand, SparsePO trades-off reward values for KL - meaning that when the model diverges from the reference model, **SparsePO is still able to generate more positive reviews compared to other methods.** This underlines the desired behavior of our method–by allowing more tokens to diverge from the reference we enable the policy to generalize to new answers. As mentioned in a previous response, this reward and divergence should be used to analyse the behavior of the approach, and not necessarily to evaluate its efficacy.
> >
> > In the statement I highlighted in your reponse, could you point me to your experimental results for that? It is indeed good if SparsePO does that even at high KL divergence compared to other methods. Could you highlight this in your paper?
> >
> > >> [Q2] The comparisons we make for KL divergence are relative and not based on hard thresholds. We observe that TDPO-v1 overall does achieve lower KL compared to SimPO, DPOP and SparsePO, bringing to a middle point in regards with the other reported PO algorithms.
> >
> > I am not suggesting you should base your analysis on hard thresholds. I am suggesting that you could describe your results more quantatitvely so that your point that "TDPO-v1 overall does achieve lower KL compared to SimPO, DPOP and SparsePO" is solidly supported.
> >
> > >> [Q3] Thank you for pointing this out. This is not phrased in the best possible way, as we did not mean to claim any causal relation. We will rephrase to stress the correlation between KL divergence and obtained expected reward instead.
> >
> > Thanks. Could you provide your rephrased analysis?

---

> ### Author Response · Authors · 2024-11-28
> **Author Response**
>
> Thank you for taking the time to look at our responses. With respect to the points you raised now, please see below.
>
> > In the statement I highlighted in your reponse, could you point me to your experimental results for that? It is indeed good if SparsePO does that even at high KL divergence compared to other methods. Could you highlight this in your paper?
>
> We have added an analysis in the amended paper, Appendix C.1, lines 1049-1069. In summary, we show that responses generated by SparsePO obtain a high ground-truth reward (as given by a sentiment classifier) whilst exhibiting higher KL divergence levels, compared to PO baselines.
>
> > I am not suggesting you should base your analysis on hard thresholds. I am suggesting that you could describe your results more quantitatively so that your point that "TDPO-v1 overall does achieve lower KL compared to SimPO, DPOP and SparsePO" is solidly supported.
>
> > Thanks. Could you provide your rephrased analysis?
>
> Apologies for the misunderstanding, we have now included quantitative comparisons in our analysis (please see lines 232-244 in the updated manuscript).
>
> We have amended this in the newest version of the paper that is now uploaded. Please refer to lines 232-244 for a re-phrasal.
>
> > Thanks for this and so empirical results suggest that lower sparsity leads to higher KL and Rewards. I would have expected based on this analysis, the paper would attempt to find good performing models at lower KL than other PO methods. Because otherwise the masks are not sparse (sparsity\% < 20\%) as suggested by the provided Table.
>
> Thank you for this remark, please let us clarify.
>     We investigated two SparsePO setups, one in which a common mask is learned for both U and D (SparsePO$[m_u=m_d]$) and another in which a separate mask in learned for each (SparsePO$[m_u \neq m_d]$).
>     For $m_u=m_d$, it is indeed the case that a low sparsity leads to higher KL and rewards (reported results in the response table).
>     However, for $m_u \neq m_d$, the sparsity in $m_u$ and in $m_d$ both interplay with KL, as seen in Figure 3 of the paper.
>     Notably, high $\beta$ and high $m_u$ sparsity leads to low KL, whilst low $\beta$ and high sparsity leads to high KL.
>     This last setup shows that SparsePO$[m_u \neq m_d]$ achieves high reward mask ($m_u$) sparsity, low KL mask ($m_d$) sparsity, high KL divergence and is still performant.
>     This result also confirms insights in previous works investigating sparsity in the objective's reward term \citep{yang2024selective}.
>
> > Could you point me to specific systems in your results that have non-trivial level of sparsity and with substantial gains compared to other PO methods? I think these should be highlighted, but are not clear in the current version.
>
> We provide sparsity plots for sentiment control (IMDB) and text-to-code generation (MBPP) in Section 3.2 and 3.5, respectively. As we observe, the sparsity of the masks changes very differently across tasks throughout training. In IMDB, we start with very high sparsity (80\%) which eventually decreases whereas in code sparsity is overall stable between 20-26\%. These results indicate that the amount of sparsity of each task significantly varies, making it `not trivial'. We also provide a complementary analysis, looking at the distribution of mask values for HH and TL;DR in Appendix C.3, which combined with our downstream task results suggest that sparsity does help achieve better performance.
>
> > Why can't other PO methods such as DPO achieve this by using a smaller beta value? And I am not sure what results in the paper specifically support the claim.
>
> Please refer to our additional analysis in Appendix C.1, where we show that for a low value of beta ($0.1$), DPO and TDPO v1 are not able to achieve high rewards for high KL values.
>
> > What is the reference performance that the +10\% is measured? It will be helpful to put together a table showing the best performing model produced by each method and controlled for sampling temperature. From my reading, the claim is based on comparing SparsePO to SFT rather than to DPO.
>
> We apologize for the confusion, the win-rates of HH were accidentally mixed with the ones reported for TL;DR. We have updated the tables and we also present the correct results in Figures 5 and 7 of the latest version of the paper. Based on the highest score of each PO method, for HH we get: +6.8\% over TDPO-v1, +12.6\% over TDPO-v2 and +5.6\% over DPO with SparsePO.
> For TL;DR we get: +1.8\% over TDPO-v1, +4.6\% over TDPO-v2 and +3.2\% over DPO with MaPO.

---

### Official Review · Reviewer_6TmP · 2024-10-30

**Soundness:** 3
**Presentation:** 2
**Contribution:** 2
**Rating:** 5
**Confidence:** 4

**Summary:**

The paper introduces Sparse Preference Optimization (SparsePO), a novel approach for controlling large language models' preference alignment through sparse token masks. The authors note that current preference optimization methods (like DPO) treat all token weights equally, whereas human preferences often depend on specific words or phrases. SparsePO introduces flexible weight masks, enabling models to automatically learn weights for KL divergence and rewards during training, thereby improving adaptation to human preferences.

**Strengths:**

1. SparsePO introduces dynamic token weighting, enhancing model adaptability and generation diversity across different preference criteria.
2. The method is evaluated across multiple datasets.

**Weaknesses:**

1. The core motivation - that human preferences depend on specific words rather than equally on all tokens - lacks empirical and theoretical validation.
2.  The introduction of m(y_t) in Equation 3 does not guarantee optimization equivalence with previous work (Zeng et al., 2024).
3.  The learnable sparse mask implementation using a single feed-forward network requires more theoretical justification. Additionally, the method's sensitivity to model architecture and data distribution needs further examination, particularly for implicit alignment tokens.
4.  Section 3.3's evaluation focuses on reasoning tasks while omitting crucial assessments of helpfulness and harmlessness metrics, diminishing the significance of minimal performance degradation in reasoning.
5.  Section 3.4's summarization task evaluation would benefit from GPT-4-based win rate metrics. The improvements are marginal, with some metrics underperforming existing baselines.
6.  Section 3.5's experiments suffer from insufficient training data and show inferior performance compared to standard DPO, potentially contradicting the paper's claims.

**Questions:**

Please refer to the weakness part.

---

> ### Author Response · Authors · 2024-11-18
> **Author Response**
>
> Thank you for your time and feedback on our work.
>
> - [W1] Thank you for your comments. Regarding validation for the core motivation of our work, we state the
> following. From a reinforcement learning perspective, Knox et al. [2022] lay down theoretical foundation
> on why taking into account the preference during each transition in a trajectory segment (e.g. explicitly
> modeling the reward of each part of a response) is better suited for human preference than modeling
> preference over the complete segment, a.k.a outcome reward modeling. This line of work has lately taken
> shape into ‘process’ reward modeling [Lightman et al., Wang et al., 2024], which has proven empirically
> beneficial for tasks in which the response can be divided into multiple steps, e.g. multi-step reasoning
> tasks. For the specific case of modeling tokens as individual steps, Zeng et al. [2024] further build on this
> line of work and laid theoretical and empirical validation, in which this work also builds on. Other work
> has also reported evidence of the importance of token-level modeling at different stages of training such
> as pretraining [Lin et al., 2024] and preference optimization [Yang et al., 2024]. From a decision making
> perspective, as studied in cognitive psychology and economics, human preference can be more precisely
> explained in realistic scenarios through dynamic decision making [Hotaling et al., 2015], in which a series
> of interdependent actions must be taken over time to achieve a goal. In such scenarios, preference is
> modeled at each time step, influenced by previous choices. In the context of natural language tasks, the
> need for non-uniform preference at the token level can become intuitive, e.g. particular words rendering
> a complete response toxic or positively sentiment laden. Figure 1 presented an example of such a case,
> where DPO implicitly learned preference at the sub-response level. In contrast, for formal language
> tasks such as code generation, such intuition is less clearly defined, since all tokens contribute to the
> executability of a function. Please refer to the General Response for more elaboration on this. Lastly,
> we would like to state that we will gladly present any additional validation you deem necessary, should
> you ask for it.
>
> - [W2] We fear there might be a misunderstanding, we will try to make the wording in the paper clearer. We do
> not claim optimization equivalence with the TDPO paper, we simply follow a similar process to derive
> the final objective, as presented in Equation (6). Please check the Appendix for the detailed derivation
> steps. Our objective is equivalent to TDPO only when both mu and md masks are equal to 1.0 (as
> stated in lines 317-319).
>
> - [W3] We understand that the choice of modeling the masks with a simple FFN might seem simplistic. We
> did some preliminary experiments with different architectures (e.g. 2 FFNs) but they exhibited worse
> performance overall.
> However, our goal in this work is to showcase the flexibility that our approach introduces in controlling
> the contribution of each token in the objective and not to investigate all potential masking strategies or
> architectures. We propose two different ways of modeling masks, first via model activations and second
> via a learnable component, but it is possible to learn a masking strategy using other methods, e.g. via
> external models or even manually creating masks for each example. We leave this exploration for future
> work.
>
> - [W4] Unfortunately, we fail to understand how to calculate helpfulness and harmlessness metrics without having
> human judgments. Could you please elaborate? To circumvent this, we have provided win-rates with
> GPT-4 as a judge as the closest evaluation instead. Please refer to the general response for the numbers.
>
> - [W5] Thank you for the suggestion, we are providing the corresponding results in the general response.
>
> - [W6] As mentioned in previous responses, we acknowledge that performance on text-to-code is not ideal.
> However, we do not expect the method to work perfectly for everything and we provide a plausible
> explanation on why we believe our method is not as effective in this particular domain in lines 428-446
> of the paper with additional arguments in the general response.

---

> ### Author Response · Authors · 2024-11-28
> **Further Clarifications needed?**
>
> Thank you again for your valuable feedback on our submission! We have attempted to respond to your comments in our rebuttal and would like to kindly ask you if you feel we addressed your concerns or if anything needs further clarification.

---

> > ### Comment · Reviewer_6TmP · 2024-12-03
> >
> > Thanks for the authors' efforts. I am willing to raise the score to 5, but I cannot give a higher score mainly because the experimental results are not particularly strong (underperforming existing methods in many scenarios like text-to-code, and summarization).

---

> > > ### Author Response · Authors · 2024-12-03
> > > **Author Response**
> > >
> > > Thank you for increasing your score, we certainly appreciate it.
> > > We would like to direct you again to the general response where we have re-framed the text-to-code experiment as a special case of SparsePO. In addition, regarding the summarization task, we do get better win-rates with our proposed weighting scheme MaPO. As mentioned in other responses, the weighting scheme that works best depends on the target task, but weighing tokens appropriately (which is the thesis of the paper) consistently helps.

---

### Official Review · Reviewer_5gHZ · 2024-11-01

**Soundness:** 3
**Presentation:** 4
**Contribution:** 3
**Rating:** 6
**Confidence:** 4

**Summary:**

Previous studies have shown that specific tokens play a key role in learning desired behaviors during pre-training and preference optimization, especially in domains where preference depends on certain aspects or subsequences. Consequently, the authors introduce SparsePO, a method for sparse token-level preference optimization. This approach aims to learn sparse masks over token-level rewards and KL divergences during training. SparsePO offers flexibility, does not rely on external models, and can be combined with different masking methods. The authors also analyze the sparsity of induced masks and their relation to KL divergence, and demonstrate quantitative and qualitative improvements when applying SparsePO in various domains with preference indicators.

**Strengths:**

1. The paper presents a technically sound approach. The motivation for proposing the objective function of SparsePO lies in the classic problem of token contribution allocation in reinforcement learning. The transformation is well-motivated and follows a logical progression.
 2. The use of masks to control the contribution of each token is a valid approach. The two proposed mask computation strategies, MAPO and SPARSEPO, are clearly described and seem feasible. The technical details provided in the methodology section, such as the equations for calculating the masks and the optimal policy, are sufficient to understand the implementation.
 3. In the experiments, the evaluation metrics used are appropriate for the tasks considered (sentiment control, dialogue, summarization, and text-to-code generation). The analysis of the trade-offs between reward and KL divergence, as well as the sparsity of the masks, provides a comprehensive understanding of the behavior of the proposed method.

**Weaknesses:**

1. Learned sparse masks do not necessarily match human preferences: In the learnable sparse mask, the author only illustrated in the paper how to adjust parameters to ensure the learned mask is sparse. However, it cannot be guaranteed that the crucial tokens are learned correctly. For example, in Figure 9(a), SparsePO-common rewards assigns almost equal rewards to all tokens.
 2. Inconsistent performance across metrics: Table 2 shows that SparsePO gains over pass@100 but has a slight decay in the remaining metrics. This indicates that while it may improve one aspect of code generation performance, it may not be uniformly beneficial across all evaluation criteria.
3. Difficulty in identifying important tokens across domains: In the code domain, especially for code execution, it is challenging to identify which particular tokens are more responsible for a program executing correctly. This is indicated by low mask sparsity levels, suggesting that the method may not be as effective in precisely weighting tokens for code generation as it could be for other domains.

**Questions:**

1. Have you attempted to train the sparse mask using the Gumbel-Softmax function?
2. When the KL constraint is very small, a larger reward value obtained by the model usually implies a higher probability of reward hacking. Can you provide some case studies to illustrate the impact of SparsePO on preventing reward hacking?

---

> ### Author Response · Authors · 2024-11-18
> **Author Response**
>
> Thank you for your time in reviewing the manuscript and for your suggestions.
>
> - [W1] This is a quite insightful remark, thank you. Firstly, we should underline that we do not actually have
> access to token-level preference annotations and as such, we cannot make any claims that SparsePO
> matches human-level preference on tokens. What we claim instead, is that by adjusting the weighing
> over tokens we can better match human-level preference on the complete response. However, we could
> provide an analysis investigating the correlation between mask values and token ‘relevance’ to a preference
> proxy, further down during the rebuttal period. Regarding our qualitative analysis of chosen and rejected
> responses (Figure 4 and 9, respectively), we do observe that the masks are sparse and indeed only tokens
> that are important towards the preference are highlighted, though we do acknowledge that the coloring
> visualization makes distinguish relative differences difficult. To further support this analysis, we provide
> statistics of the distribution of mask values in the table below, showing quantiles at 0.05 (Q5) and 0.95
> (Q95) as well as the mean (m) of the U term in Equation (6), for chosen responses. We will include
> this analysis as a histogram in the revised version of the submission. Finally, indeed, our method relies
> on the PO dataset at hand to model token-level supervision signals for ‘crucial tokens’.
>
> | Model	        |  $u_{w}$--Q5	    | $u_w$--m	| $u_w$--Q95	 | $d_w$--Q5	| $d_w$--m |	$d_w$--Q95 |
> |---------------|---------------|-----------|------------|----------|---------|---------------|
> | SparsePO$[m_u \neq m_d]$	| 0.00000	    | 0.021247	 | 0.134033	| 0.00000 | 0.147135 |	0.404297 |
> | SparsePO$[m_u=m_d]$	| 0.06481	    | 0.271115	 | 0.494629	| 0.06481 |	0.271115	| 0.494629 |
>
> - [W2 & W3] We acknowledge the low performance of our method on text-to-code generation benchmarks. We believe
> that this particular domain might not be the best for SparsePO for the reasons highlighted in lines 428-
> 445 of the paper and the general response. However, we do manage to achieve better performance based
> on the respective metrics on all the other benchmarks.
>
> - [Q1] We have not tried to train the sparse mask via Gumbel softmax, though we agree that it is a valid
> option for achieving sparsity. In our approach, we managed to regulate sparsity via L1-regularization and
> consider different avenues as beyond the scope of our work.
>
> - [Q2] Thank you for raising this concern. We delved further into the following indicative of reward hacking,
> reward ’accuracy’ (chosen reward > rejected reward), and report its progression during training for DPO,
> MaPO, and SparsePO models, in the Table below, for the sentiment control scenario. We observe that
> DPO quickly reaches a high accuracy, likely due to lowering the probability of rejected responses instead
> of increasing the probability of chosen responses. DPO’s accuracy of almost 100% is a clear indicative of
> reward hacking. For MaPO, we observe similar trends, albeit converging to a lower accuracy. However,
> SparsePO avoids reward hacking by managing to converge slower and to lower accuracies than the other
> models, whilst remaining performant in downstream tasks, as evidenced by our main results.
>
> | Step | DPO   | MaPO  | SparsePO$[m_u=m_d]$ | SparsePO$[m_u\neq m_d]$ |
> |------|-------|-------|-------|-------|
> | 0    | 0     | 0     | 0     | 0     |
> | 100  | 74.37 | 77.19 | 60.94 | 32.50 |
> | 200  | 94.69 | 90.94 | 85.31 | 42.50 |
> | 300  | 98.44 | 90.94 | 84.38 | 52.19 |
> | 400  | 99.69 | 97.19 | 72.81 | 50.31 |

---

> ### Author Response · Authors · 2024-11-28
> **Further clarifications needed?**
>
> Thank you again for your valuable feedback on our submission! We have attempted to respond to your comments in our rebuttal and would like to kindly ask you if you feel we addressed your concerns or if anything needs further clarification.
>
> Additionally, we added further evidence to the inquired points in the paper, please have a look at the latest version where additions/changes are highlighted with different colors. Specifically;
> > Learned sparse masks do not necessarily match human preferences: ...
>
> [response] This is a quite insightful remark, thank you. Firstly, we should underline ...
>
> We also added an analysis on the distribution of mask values and token-level KL divergence in Appendix C.3., for the tasks of summarization, dialogue, and code generation.
> In summary, we show that in cases where mask values highly concentrate around zero (and hence, high sparsity), KL divergence tends to be high throughout training, indicating that the few tokens that are allowed to diverge are diverging quite largely. Please also refer to Appendix C.6. for additional qualitative analysis on different tasks and the tokens that are crucial for them.
>
> > When the KL constraint is very small, a larger reward ...
>
> [response] Thank you for raising this concern. We delved further ...
>
> Summarization is another example where SparsePO avoids reward hacking, as illustrated in Appendix C.3.
> In summary, we find that most tokens in a summary response contribute to the reward whilst the KL divergence is higher than baselines.

---

### Official Review · Reviewer_JkpM · 2024-11-04

**Soundness:** 3
**Presentation:** 3
**Contribution:** 3
**Rating:** 6
**Confidence:** 3

**Summary:**

The paper proposes SparsePO, a new approach for Preference Optimization (PO) with a token-level focus. Specifically, SparsePO uses sparse token masks to assign different weights to specific tokens, allowing flexible optimization. SparsePO uses dynamic or model-derived masking strategies.
Quantitative and qualitative experiments on a few varied tasks show that SparsePO works well and provides improvements in some of the cases.

**Strengths:**

1. The paper is clear and well-written.
2. The method seems novel, has a clear and well-established motivation, and is mathematically rigorous.
3. The paper performs experiments on a varied set of tasks.
4. The sentiment control experiments show good trade-offs between KL divergence and reward. The "Sparsity and Token-level KL divergence" experiment is insightful.
5. Nice improvements on IFEVAL and BBH with H&H training.

**Weaknesses:**

1. The  TL;DR dataset can be unfaithful and a small set of 120 prompts can hinder results further. Hence I tend to suspect the results. This is a more experimental design problem than a method problem. For faithfulness, AFAIK there are better methods like Q^2, True, GPM, and more.
2. Although the H&H shows some nice results, the size of the model combined with the difficulty of the benchmarks (OpenLLM-2 is designed to be much harder than 1) limit the ability to properly assess the method capabilities. Again this is more on the experimental design side. Running on the version may be better. Also, considering stronger models in the 1B range can also help (See Phi, SmolLM, Qwen2; see this blog reporting small LLMs results on OLLM-V.1 https://huggingface.co/blog/smollm).
3. The results on text-to-code are mixed or even negative raising questions regarding the versatility of the method.

**Questions:**

1. Although PPO is less used, many recent papers show that it is not inferior to DPO, so is it justified to exclude this from the evaluation?

Comments:
1. There are a few different terminology regarding SparsePO, including method, strategy, objective, and framework. This can be confusing to the reader.

---

> ### Author Response · Authors · 2024-11-18
> **Author Response**
>
> We would like to thank you for your time and insightful comments!
>
> - [W1] We understand that TL;DR can be problematic as a dataset and as such, to avoid any possible bias based on summary length, in the paper we filtered out training instances with chosen and rejected responses with a length difference greater than 100 words. From this resulting filtered dataset, we uniformly sampled 40k and 8k preference instances to form the training and test set, respectively (more information about this can be found in the Appendix). From this test set of 8k, we further uniformly sampled 100 unique prompts. We will extend this evaluation to include all the 1889 unique prompts of the test set. We will add these results in our rebuttal in the coming days.
> Regarding faithfulness, could you please elaborate on what analysis you would like to see? We can suggest to further filter out instances in which the gold summary is deemed as unfaithful, based on an existing faithfulness metric. Or to separate results of automatic metrics by faithfulness level of the system summaries. e.g. separated by low, mid, and high faithfulness. Regarding the suggested metrics we could report ANLI and SummaCSZ as well. Unfortunately, we cannot include Q2 easily, as no official implementation is available.
> - [W2] We agree that OpenLLM-v1 might be a better choice for the model size that we report, hence we will
> also share the evaluation results on OLLM-v1 by the end of the discussion period. In the meantime we
> have expanded our HH analysis to include win-rates; please refer to the General response.
> - [W3] Indeed we observe the results in text-to-code generation might not always be positive. We attribute this
> to the observation that for successful code execution, restricting learning to only a handful of tokens
> might not be adequate as mentioned in the general response and in lines 428-446.
> - [Q1] Following previous work on offline preference optimization, in this study we chose to focus on offline
> algorithms and leave online ones to future work. Nevertheless, we certainly agree that PPO is not
> inferior to DPO, as prior work has shown. Actually, our approach is orthogonal to PPO, meaning that it
> can be easily extended to include masks.
> - [C1] Thank you for pointing these inconsistencies (method/stratefy/objective/framework), we will make sure to address them in the manuscript.

---

> ### Comment · Reviewer_JkpM · 2024-11-21
> **Review Response**
>
> [W1] Your analysis suggestion of filtering unfaithful data (as done here: https://aclanthology.org/2023.findings-acl.220.pdf) indeed makes sense and will increase my trust in your data and results. I think ANLI and SummaCSZ are sufficient.
>
> [W2] I await your new results. I am unsure how you got the new HH win rates, and you got the 10%/3% improvement.
>
> [W3] I find your response problematic. You state that your method is more capable in use cases that aim to "reduce toxicity" and "improve style", but less capable in code as all words are important in tasks that seek to "improve functional correctness".  Given that statement, it is odd that you train on HH and evaluate on OpenLLM-2 which aims to provide a broad set of evaluations that don't relate to toxicity and style. Moreover, I think that math is very similar to code in that aspect, but yet training in HH improved it a bit. Still, you can suggest that training on the target task is different. So probably to convince me that your explanation is not just an ad-hock explanation of a bad result but a real failure mode of the approach it will be beneficial to show it has some predictive power. For example, you can train on math tasks and demonstrate lower results with your method.

---

> > ### Author Response · Authors · 2024-11-28
> > **Author Response**
> >
> > Thank you for replying to our comments!
> >
> > - [W1] We have updated the corresponding results over filtered data in Figure 6, after filtering the instances based on the degree of faithfulness of the reference summary wrt the document using SummaC$_{ZS}$. For completeness, we provide the same metric results over the entire test set in Appendix C.5.
> >
> > - [W2] The process we followed to obtain the win-rates is described in the general response. We accidentally reported the TL;DR rates instead of the HH ones. Please refer to the updated tables for the results, or the newest version of our manuscript (Figures 5 and 7).
> > Results in OLLM-v1 can be found below. We indeed find that we are able to get better performance to SFT with v1. We also report these numbers in Appendix C.4 of the latest version of the paper.
> >
> > | Model                   | ARC | HellaSwag | TruthfulQA | MMLU | Winogrande | Avg. |
> > |-------------------------|-----|-----------|------------|------|------------|------|
> > | SFT                     | 26.52 | 46.74 | 41.63 | 22.49 | 56.43 | 38.76 |
> > | DPO                     | 27.61 | 47.64 | 42.35 | 23.87 | 56.80 | 39.65 |
> > | TDPO v1                 | 30.20 | 49.05 | 41.35 | 24.11 | 56.09 | 40.16 |
> > | TDPO v2                 | 28.95 | 48.61 | 43.14 | 23.48 | 56.27 | 40.09 |
> > | SimPO                   | 28.50 | 33.07 | 47.73 | 23.21 | 51.93 | 36.88 |
> > | DPOP                    | 30.38 | 47.91 | 43.48 | 22.83 | 56.09 | 40.13 |
> > | MaPO                    | 29.10 | 50.89 | 41.63 | 24.63 | 57.77 | 40.80 |
> > | SparsePO$[m_u=m_d]$     | 28.73 | 48.48 | 42.23 | 24.91 | 59.12 | 40.69 |
> > | SparsePO$[m_u \neq m_d]$| 29.92 | 47.15 | 42.97 | 23.64 | 57.46 | 40.22 |
> >
> > - [W3] Thanks for this insightful comment. Indeed, we agree that mathematical reasoning and code generation are similar in terms of the `correctness' requirement of the response, however with the following crucial difference.
> > In current math benchmarks (e.g. MATH) correctness is evaluated as obtaining the correct final answer, regardless of the correctness of intermediate reasoning steps.
> > Hence, a model has more liberty in generating a response consisting of steps and the final answer, i.e. if a response contains incorrect intermediate steps but the correct final answer, it will be deemed as correct.
> > However, in our text-to-code setup, it is crucial that the response not only executes but also that it returns the correct answer for all test units.
> > In this case, an incorrect intermediate logical step in the response, even if executable, will prompt an incorrect answer (or fail to run).
> >
> > With that cleared, we claim that our method works better on tasks that rely on natural language, where some words are naturally more important for preference than others. This includes the standard Preference Optimization goals of reducing toxicity or style adaptation, but it extends on reasoning tasks as well when that reasoning is happening through natural language.
> > That is the case in MATH as well, since performance there is enabled by Natural Language instruction through chain-of-thought reasoning. As such, we do not find it surprising that SparsePO leads to improvements there as well.
> > Please refer to Appendix C.6 for a qualitative example showcasing this scenario.
> >
> > Regarding the experimental setup in HH, we indeed agree that more targeted evaluation was needed and as such, we added win-rates, as mentioned before, as well as results over TruthfulQA, included in the OLLM-v1 results in Appendix C.4.

---

> > > ### Comment · Reviewer_JkpM · 2024-11-28
> > >
> > > Thank you for your response.
> > > [W1] Figure 6 is small, and 7 does not have a legend of its own making it hard to read.
> > > [W2] Results on OLLM-v1 make more sense. Are they significant?
> > > Given the three versions of your method, the win rate seems mixed, and it is not clear which one is better, and if there is one that is better across the different experiments. SparsePO[mu = md] in HH, AND Mapo in TL;DR.
> > > [W3] Okay. Nonetheless, you did not show that your explanation has a predictive power.
> > >
> > > The experiments do not match the expectations raised by the motivation, focusing on reducing toxicity or style adaptation.
> > > Improvements are relatively small even for natural language tasks, and not systematic across the methods versions, and the different temperatures, making it hard to determine the nature of the improvement without significant testing.

---

> ### Author Response · Authors · 2024-12-02
> **Author Response**
>
> Thank you for your comments.
>
> > Figure 6 is small, and 7 does not have a legend of its own making it hard to read.
>
> We will make sure to add a legend to Figure 7. Since we cannot upload another paper version, please use any other legend (e.g. in Figure 6) as we made sure that the colors remain consistent across methods.
>
> > Results on OLLM-v1 make more sense. Are they significant?
>
> All automatic metric results (in HH and TL;DR) are tested for statistical significance at the system level using Bootstrap resampling (Davison \& Hinkley, 1997) with a 95\% confidence interval (Appendix B.3, L949-L951).
>  Such non-parametric test is most appropriate when the distribution of a metric value cannot be assumed to be Gaussian, as is the case for metrics employed in NLG tasks. Additionally, Bootstrap resampling has become the established method to report the significance of results in tasks such as summarization and QA.
>
> > Given the three versions of your method, the win rate seems mixed, and it is not clear which one is better, and if there is one that is better across the different experiments. SparsePO[mu = md] in HH, AND Mapo in TL;DR.
>
> In the paper we have proposed 3 different strategies for applying masks over tokens to control the contribution of each token in the PO objective.
> Across experiments, the methodology itself clearly shows a consistent improvement in cases where indeed only a handful of tokens are important for steering the preference. As expected, which specific masking strategy is most effective ultimately depends on the preference proxy.
>
> > Okay. Nonetheless, you did not show that your explanation has a predictive power.
>
> In the paper we report several cases that illustrate the predictive power of our method.
> In sentiment control, the Pareto frontier and our additional analysis in the previous round of responses illustrate that our method is able to generate more samples with the desired proxy.
> In summarization, existing metrics for summary quality and gpt-4 win-rates illustrate gains over other PO methods.
> In HH, OpenLLM v1 and v2 scores with gpt-4 win-rates provide evidence that our method improves over baselines.
> Could it be possible to clarify what do you mean by predictive power if the above response is not adequate?
>
> > The experiments do not match the expectations raised by the motivation, focusing on reducing toxicity or style adaptation.
>
> Although we used some examples from HH and style adaptation to help illustrate our motivation, please note that by no means is our motivation limited to those tasks.
>     Our motivation is that preference can be better attributed by weighting different tokens in a response, especially in natural language, an insight backed by previous work.
>     Most notably, we presented evidence that our methodology leads to gains across challenging benchmarks beyond toxicity or style adaptation (OLLM v1 and v2).
>
> > Improvements are relatively small even for natural language tasks, and not systematic across the methods versions, and the different temperatures, making it hard to determine the nature of the improvement without significant testing.
>
> We would like to point out that the relatively small difference in automatic metric values might be due to their inherent lack of sensitivity and their difficulty.
> For instance, a difference of half a percentage point (in average) over the OLLM leaderboard tasks is considered as a significant improvement in previous PO work.
> Additionally, as mentioned above, please note that all our results are tested for statistical significance.
> Regarding the observation of systematic differences across temperatures, it is a commonly observed behavior of PO methods to exhibit wider performance gaps at lower temperatures, with the gap completely closing at $T=\{0.75,1.0\}$.
> Our method exhibits this behavior too.
> Finally, the effectiveness of our methodology is reported across a varied set of tasks and discussed at length with dedicated experiments, as others reviewers kindly acknowledged.

---

### Author Response · Authors · 2024-11-18
**General Response to Reviewers**

We would like to thank all reviewers for their time in reviewing the manuscript and providing their insightful
comments. Here, we will be addressing comments that are common among reviewers and provide some
additional results to support the paper’s thesis. We are also currently performing additional experiments to
address reviewers concerns; we will be adding these results as they become available.
To further support SparsePO’s efficacy compared to related work and concerns raised by multiple reviewers, we
provide win-rates for the HH and TL;DR benchmarks using GPT-4 (gpt-4-turbo) as the judge on 100 randomly
chosen prompts of the test set. In particular, we compare the gold chosen response with a response generated
by our model after PO training and ask GPT-4 to select the best response for various temperatures. The
prompts used follow Rafailov et al. [2023] and Amini et al. [2024] (we will also include those in an updated
version of the paper by the end of the response period). As observed from the tables below, in HH, SparsePO-
common obtains the best win-rates across the board with SparsePO-indp being the close second. For TL;DR,
systems are close with MaPO showing the highest rates.
Given that multiple reviewers raised that the benefits of our proposed method were unclear, we summarize
them below. Please note that these benefits take into account the additional results we provide.

With respect to the paper’s contributions:
- Flexibility in increasing or suppressing the contribution of each token during preference optimization.
- Our method produces a non-trivial level of sparsity that confirms, alongside the reported performance
gains, that not all tokens are required for effective preference optimization.
- Using token-level masks, models are injected with information that allow them to recognize which tokens
are crucial for the target proxy, potentially enabling external signals as well. Here, we simply focused on
either using the reference model activations or learning a mask via learned parameters.
- Our method is able to produce higher rewards at larger KL divergence values enabling models to generate
more diverse responses.
- We are able to achieve improved performance in terms of gpt-4 evaluations on HH and TL;DR bench-
marks.

Furthermore, our analysis has shown:
- Performance-wise SparsePO/MaPO have achieved: +0.91 pp on OpenLLM Leaderboard-v2, overall with
+1.32 and +2.03 specifically on instruction following benchmarks.
- +10% winrates in Anthropic-HH single-turn dialogue (using GPT-4-turbo as a judge across different
temperatures).
- +3% winrates in TL;DR summarization (using GPT-4-turbo as a judge across different temperatures).
With respect to the results on Text-to-code generation, we aim to re-frame it as negative case for our method,
where a sparse approach cannot work due to the nature of the task, hence, wee provide the following intuitions.

The thesis behind SparsePO is that human preference is often reliant on specific words or phrases in sentences,
which is well observed and supported in Natural Language. However, code sequences are heavily structured
and every ‘word’ is intricately reliant on all other ‘words’ in the sequence, i.e. there is little information that
may be considered redundant. As such, a weighing scheme will effectively ignore parts of the sequence that
can be crucial. Additionally, in natural language PO is mostly used to reduce toxicity or improve style, but
in code it is strictly to improve functional correctness, and ignoring any ‘word’ in a code sequence will most
certainly lead to a functionally incorrect solution.

---

> ### Author Response · Authors · 2024-11-18
> **Win rates**
>
> |model                   |Temperature|HH winrate|
> |------------------------|-----------|-------|
> |SFT                     |0          |0.148  |
> |SFT                     |0.25       |0.12   |
> |SFT                     |0.5        |0.16   |
> |SFT                     |0.75       |0.224  |
> |SFT                     |1          |0.2    |
> |DPO                     |0          |0.25   |
> |DPO                     |0.25       |0.286  |
> |DPO                     |0.5        |0.344  |
> |DPO                     |0.75       |0.318  |
> |DPO                     |1          |0.34   |
> |TDPO v1                 |0          |0.302  |
> |TDPO v1                 |0.25       |0.292  |
> |TDPO v1                 |0.5        |0.306  |
> |TDPO v1                 |0.75       |0.308  |
> |TDPO v1                 |1          |0.34   |
> |TDPO v2                 |0          |0.15   |
> |TDPO v2                 |0.25       |0.182  |
> |TDPO v2                 |0.5        |0.214  |
> |TDPO v2                 |0.75       |0.276  |
> |TDPO v2                 |1          |0.27   |
> |MaPO                    |0          |0.294  |
> |MaPO                    |0.25       |0.3    |
> |MaPO                    |0.5        |0.318  |
> |MaPO                    |0.75       |0.304  |
> |MaPO                    |1          |0.294  |
> |SparsePO$[m_u=m_d]$     |0          |0.344  |
> |SparsePO$[m_u=m_d]$     |0.25       |0.348  |
> |SparsePO$[m_u=m_d]$     |0.5        |0.4    |
> |SparsePO$[m_u=m_d]$     |0.75       |0.402  |
> |SparsePO$[m_u=m_d]$     |1          |0.408  |
> |SparsePO$[m_u \neq m_d]$|0          |0.314  |
> |SparsePO$[m_u \neq m_d]$|0.25       |0.328  |
> |SparsePO$[m_u \neq m_d]$|0.5        |0.338  |
> |SparsePO$[m_u \neq m_d]$|0.75       |0.33   |
> |SparsePO$[m_u \neq m_d]$|1          |0.314  |
>
> | model                    | Temperature | TL;DR winrate |
> | :----------------------- | :---------- | :------ |
> | SFT                      | 0           | 0.444   |
> | SFT                      | 0.25        | 0.456   |
> | SFT                      | 0.5         | 0.448   |
> | SFT                      | 0.75        | 0.386   |
> | SFT                      | 1           | 0.232   |
> | DPO                      | 0           | 0.558   |
> | DPO                      | 0.25        | 0.544   |
> | DPO                      | 0.5         | 0.492   |
> | DPO                      | 0.75        | 0.426   |
> | DPO                      | 1           | 0.286   |
> | TDPO v1                  | 0           | 0.546   |
> | TDPO v1                  | 0.25        | 0.558   |
> | TDPO v1                  | 0.5         | 0.484   |
> | TDPO v1                  | 0.75        | 0.398   |
> | TDPO v1                  | 1           | 0.282   |
> | TDPO v2                  | 0           | 0.49    |
> | TDPO v2                  | 0.25        | 0.53    |
> | TDPO v2                  | 0.5         | 0.47    |
> | TDPO v2                  | 0.75        | 0.374   |
> | TDPO v2                  | 1           | 0.24    |
> | MaPO                     | 0           | 0.542   |
> | MaPO                     | 0.25        | 0.576   |
> | MaPO                     | 0.5         | 0.492   |
> | MaPO                     | 0.75        | 0.414   |
> | MaPO                     | 1           | 0.35    |
> | SparsePO$[m_u=m_d]$      | 0           | 0.526   |
> | SparsePO$[m_u=m_d]$      | 0.25        | 0.492   |
> | SparsePO$[m_u=m_d]$      | 0.5         | 0.466   |
> | SparsePO$[m_u=m_d]$      | 0.75        | 0.418   |
> | SparsePO$[m_u=m_d]$      | 1           | 0.242   |
> | SparsePO$[m_u \neq m_d]$ | 0           | 0.534   |
> | SparsePO$[m_u \neq m_d]$ | 0.25        | 0.524   |
> | SparsePO$[m_u \neq m_d]$ | 0.5         | 0.488   |
> | SparsePO$[m_u \neq m_d]$ | 0.75        | 0.416   |
> | SparsePO$[m_u \neq m_d]$ | 1           | 0.282   |

---

> > ### Author Response · Authors · 2024-11-18
> > **Author Response references**
> >
> > - Afra Amini, Tim Vieira, and Ryan Cotterell. Direct Preference Optimization with an Offset. In Lun-Wei Ku,
> > Andre Martins, and Vivek Srikumar, editors, Findings of the Association for Computational Linguistics ACL
> > 2024, pages 9954–9972, Bangkok, Thailand and virtual meeting, August 2024. Association for Computational
> > Linguistics.
> > - Jared M Hotaling, Pegah Fakhari, and Jerome R Busemeyer. Dynamic decision making. International ency-
> > clopedia of the social & behavioral sciences, 8:708–713, 2015.
> > - W Bradley Knox, Stephane Hatgis-Kessell, Serena Booth, Scott Niekum, Peter Stone, and Alessandro Allievi.
> > Models of human preference for learning reward functions. arXiv preprint arXiv:2206.02231, 2022.
> > - Hunter Lightman, Vineet Kosaraju, Yuri Burda, Harrison Edwards, Bowen Baker, Teddy Lee, Jan Leike, John
> > Schulman, Ilya Sutskever, and Karl Cobbe. Let’s verify step by step. In The Twelfth International Conference
> > on Learning Representations.
> > - Zhenghao Lin, Zhibin Gou, Yeyun Gong, Xiao Liu, Yelong Shen, Ruochen Xu, Chen Lin, Yujiu Yang, Jian Jiao,
> > Nan Duan, et al. Rho-1: Not All Tokens Are What You Need. arXiv preprint arXiv:2404.07965, 2024.
> > - Rafael Rafailov, Archit Sharma, Eric Mitchell, Christopher D Manning, Stefano Ermon, and Chelsea Finn. Direct
> > Preference Optimization: Your Language Model is Secretly a Reward Model. In Thirty-seventh Conference on
> > Neural Information Processing Systems, 2023.
> > - Peiyi Wang, Lei Li, Zhihong Shao, Runxin Xu, Damai Dai, Yifei Li, Deli Chen, Yu Wu, and Zhifang Sui.
> > Math-shepherd: Verify and reinforce llms step-by-step without human annotations. In Proceedings of the
> > 62nd Annual Meeting of the Association for Computational Linguistics (Volume 1: Long Papers), pages
> > 9426–9439, 2024
> > - Kailai Yang, Zhiwei Liu, Qianqian Xie, Jimin Huang, Erxue Min, and Sophia Ananiadou. Selective Preference
> > Optimization via Token-Level Reward Function Estimation. arXiv preprint arXiv:2408.13518, 2024.
> > - Yongcheng Zeng, Guoqing Liu, Weiyu Ma, Ning Yang, Haifeng Zhang, and Jun Wang. Token-level Direct
> > Preference Optimization. In Ruslan Salakhutdinov, Zico Kolter, Katherine Heller, Adrian Weller, Nuria
> > Oliver, Jonathan Scarlett, and Felix Berkenkamp, editors, Proceedings of the 41st International Conference
> > on Machine Learning, volume 235 of Proceedings of Machine Learning Research, pages 58348–58365. PMLR,
> > 21–27 Jul 2024

---

### Author Response · Authors · 2024-11-28
**Latest Version Updates**

We would like to thank all the reviewers for their insightful feedback that helped us significantly improve the paper.
Based on their original comments and responses, we have updated the paper to include additional experiments as well as clarifications to strengthen our work. We have highlighted the corresponding places with a different color for their convenience. We also briefly include the changes here:
- Win-rates for HH and TL;DR with GPT-4 as a judge, in Figures 5 and 7.
- Analysis in Appendix C.1 with respect to the reward and response-level KL trade-off [Reviewer hU2n]
- Updated results over TL;DR after filtering low faithfulness instances in Figure 6 and Appendix C.5 [Reviewer JkpM]
- Results on the Open LLM Leaderboard v1 in Appendix C.4 [Reviewer JkpM]
- More accurate description of the relationship between KL divergence and expected reward in the Pareto frontier in lines 232-246 [Reviewer hU2n]
- Further intuitions about text-to-code results in Section 3.5.
- Discussion about how the value of $\beta$ works in tandem with mask values in lines 260-263 and Appendix C.2 [Reviewer hU2n].
- Further analysis regarding the correlation between mask distribution and token-level KL divergence in Appendix C.3 [Reviewer hU2n]
- Our code is now available (check Appendix B.1)
- We further evaluated Pythia-1.4B on the HumanRankEval test set, results provided in Appendix C.7.
- Additional qualitative results for HH, MATH and code in Appendix C.6, Figure 22.

---

### Meta-Review · Area_Chair_nUeL · 2024-12-08

**Metareview:**

The paper proposes SparsePO, a preference optimization (PO) algorithm that aims to automatically learn to weight the KL divergence and reward corresponding to each token during PO training, motivated by the finding that human preference usually depends more on specific tokens rather than equally across all tokens. While the reviewers find the method to be interesting, they also raised several issues such as the insufficiently justified token mask designs and the marginal performance advantage over baselines.

**Additional Comments On Reviewer Discussion:**

The reviewers reached a consensus on rejection.

---

### Decision · Program_Chairs · 2025-01-22

Reject